# Regional response to light illuminance across the human hypothalamus

Islay Campbell[1†], Roya Sharifpour[1†], Jose Fermin Balda Aizpurua[1], Elise Beckers[1,2], Ilenia Paparella[1], Alexandre Berger[1,3,4], Ekaterina Koshmanova[1], Nasrin Mortazavi[1], John Read[1], Mikhail Zubkov[1], Puneet Talwar[1], Fabienne Collette[1], Siya Sherif[1], Christophe Phillips[1], Laurent Lamalle[1], Gilles Vandewalle[1]*

[1]GIGA-CRC Human Imaging, University of Liège, Liège, Belgium; [2]Faculty of Health, Medicine and Life Sciences, School for Mental Health and Neuroscience, Alzheimer Centre Limburg, Maastricht University, Maastricht, Netherlands; [3]Synergia Medical SA, Mont-Saint-Guibert, Belgium; [4]Institute of Neuroscience (IoNS), Department of Clinical Neuroscience, Université Catholique de Louvain (UCLouvain), Woluwe-Saint-Lambert, Belgium

**\*For correspondence:**
gilles.vandewalle@uliege.be

†These authors contributed equally to this work

**Competing interest:** The authors declare that no competing interests exist.

## eLife assessment

This **fundamental** work describes the complex interplay between light exposure, hypothalamic activity, and cognitive function. The evidence supporting the conclusion is **compelling** with potential therapeutic applications of light modulation. The work will be of broad interest to basic and clinical neuroscientists.

**Abstract** Light exerts multiple non-image-forming biological effects on physiology including the stimulation of alertness and cognition. However, the subcortical circuitry underlying the stimulating impact of light is not established in humans. We used 7 Tesla functional magnetic resonance imaging to assess the impact of variations in light illuminance on the regional activity of the hypothalamus while healthy young adults (N=26; 16 women; 24.3±2.9 y) were completing two auditory cognitive tasks. We find that, during both the executive and emotional tasks, higher illuminance triggered an activity increase over the posterior part of the hypothalamus, which includes part of the tuberomamillary nucleus and the posterior part of the lateral hypothalamus. In contrast, increasing illuminance evoked a decrease in activity over the anterior and ventral parts of the hypothalamus, encompassing notably the suprachiasmatic nucleus and another part of the tuberomammillary nucleus. Critically, the performance of the executive task was improved under higher illuminance and was negatively correlated with the activity of the posterior hypothalamus area. These findings reveal the distinct local dynamics of different hypothalamus regions that underlie the impact of light on cognition.

## Introduction

Light exerts multiple non-image-forming (NIF) biological effects that influence the quality of sleep and wakefulness, and higher illuminance is known to stimulate alertness and cognition (*Campbell et al., 2023*). The biological effects of light primarily rely on a subclass of retinal ganglion cells that are intrinsically photosensitive (ipRGCs) because they express the photopigment melanopsin, which is maximally sensitive to photons with wavelength ~480 nm. IpRGCs combine the light signalling of rods and cones to their intrinsic photosensitivity and, collectively, the biological effects of light present a maximal sensitivity to the shorter blue wavelength of visible light (*Do, 2019*). IpRGCs project to

multiple subcortical brain areas and their denser projections are found within the hypothalamus, particularly in nuclei involved in sleep and wakefulness regulation (*Scammell et al., 2017*; *Do, 2019*). The suprachiasmatic nucleus (SCN), which is the site of the principal circadian clock, receives the strongest inputs from ipRGC inputs, over the anterior part of the hypothalamus (*Hattar et al., 2006*). Other nuclei also receive ipRGC projections: the subparaventricular zone, one of the main output routes of the SCN, the ventrolateral preoptic nucleus (VLPO) and the preoptic nucleus (PON) involved in sleep initiation and also found in the anterior part of the hypothalamus; the lateral hypothalamus (LH), site of the orexinergic wake-promoting neurons and melanin-concentrating hormone sleep-promoting neurons, found in contrast over the lateral and posterior parts of the hypothalamus (*Scammell et al., 2017*; *Do, 2019*).

The brain circuitry underlying the biological effects of light has mostly been uncovered in nocturnal rodent models (*Campbell et al., 2023*; *Do, 2019*). Translation to diurnal human beings, where the later maturation of the cortex allows for complex cognitive processing (*Braak and Del Tredici, 2015*), remains scarce. In particular, whether hypothalamus nuclei contribute to the stimulating impact of light on cognition in humans is not established.

We addressed this question using ultra-high-field (UHF) 7 Tesla (7T) functional magnetic resonance imaging (fMRI) in healthy young adults exposed to light of various illuminance while engaged in two different auditory cognitive tasks. We find that higher illuminance increased the activity of the posterior part of the hypothalamus encompassing the mamillary bodies (MB) and parts of the LH and tuberomammillary nucleus (TMN). In contrast, higher illuminance decreased the activity over the anterior and ventral parts of the hypothalamus encapsulating notably the SCN and another part of the TMN. Critically, the pattern of modulation was consistent across the two cognitive tasks. Importantly, the performance of the complex cognitive task was improved under higher illuminance while the activity of the posterior part of the hypothalamus was correlated to task performance. The findings reveal the distinct local dynamics of different hypothalamus areas in response to changing illuminance that may contribute to light's impact on cognition.

## Results

Twenty-six healthy young adults (16 women; 24.3±2.9 y; *Supplementary file 1a*) completed two auditory cognitive tasks encompassing, respectively, the executive (*Collette et al., 2005*) and emotional (*Grandjean et al., 2005*) domains, while alternatively maintained in darkness or exposed to short periods (<1 min) of light of four different illuminances (0.16, 37, 92, 190 melanopic equivalent daylight illuminance - mel EDI- lux; *Supplementary file 1b*; *Figure 1*). The hypothalamus of each participant was segmented into five subparts – inferior-anterior, superior-anterior, inferior-tubular, superior-tubular, and posterior (*Figure 2A*) – so we could consistently extract the regional effect of illuminance change on fMRI blood-oxygen-level-dependent (BOLD) signal over most of the hypothalamus volume.

### The impact of illuminance variations on the activity of the hypothalamus is not uniform

The main analyses aimed at isolating differences in the overall impact of illuminance changes among the 5 hypothalamus subparts. For each subpart, we extracted an index of the illuminance impact as their average regression coefficients between their responses to the tasks and the illuminance levels. These analyses showed significant differences between the hypothalamus subparts for the executive (generalized linear mixed models [GLMM]; main effect of the subparts; p=0.002) and emotional (GLMM; main effect of the subparts; p<0.0001) tasks, revealing that, during both tasks, the variations in illuminance affected the activity of the 5 hypothalamus subparts differently (*Figure 2B and C*; *Table 1*). A nominal main effect of the task was detected for the emotional task (p=0.049; *Table 1*) but not for the n-back task. For both tasks, there was no significant main effect for any of the other covariates and post hoc analyses showed that the index of the illuminance impact was consistently different in the posterior hypothalamus subpart compared to the other subparts (p_corrected ≤0.05, *Table 1*). Importantly, whole-brain analyses confirmed that increasing illuminance resulted in a local increase and decrease of activity that could be detected, respectively, over the posterior and inferior subparts of the hypothalamus (*Figure 2D–G*). This shows that our results do not come from a relatively

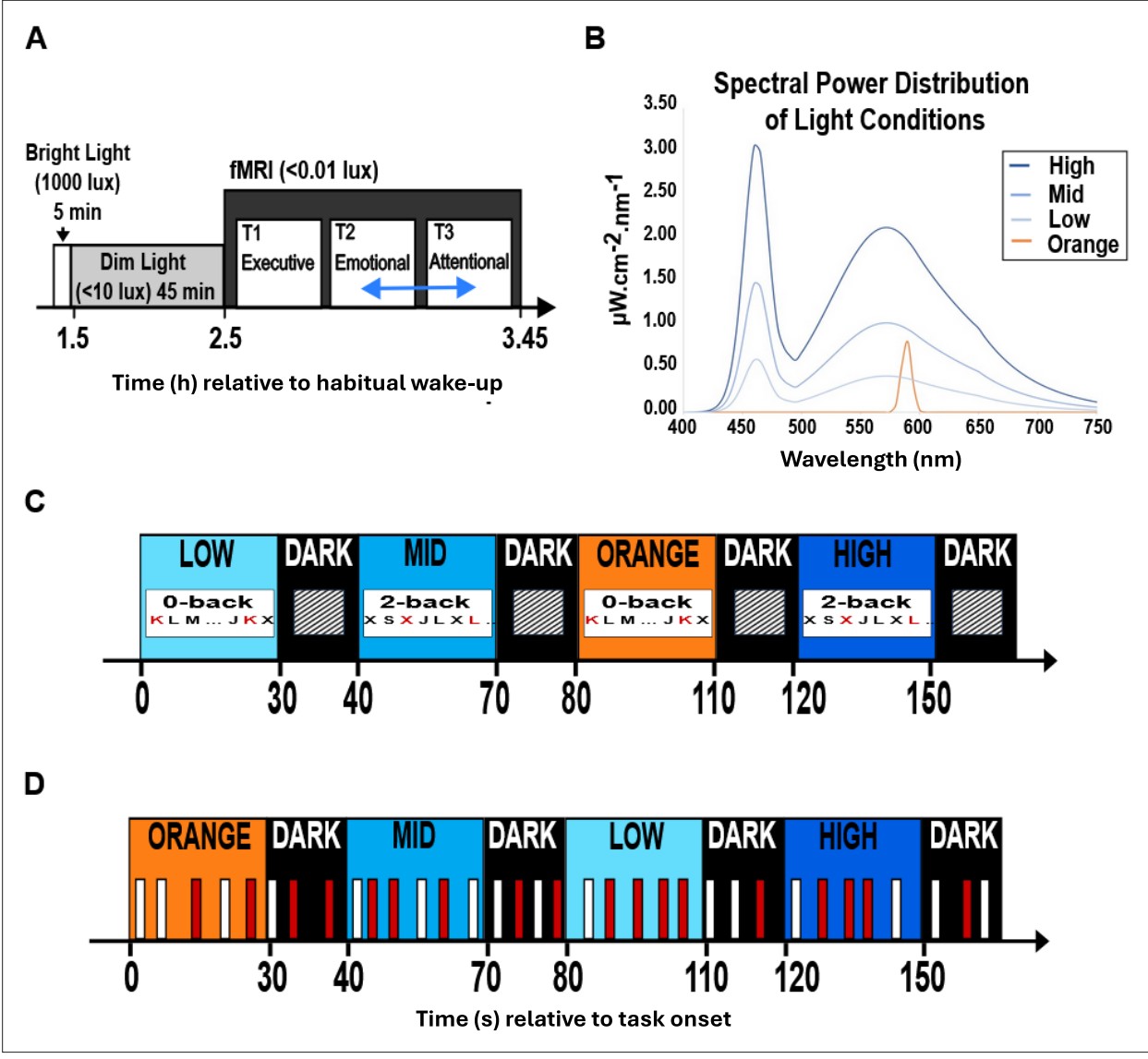

**Figure 1.** Experimental protocol. (**A**) Overall timeline. After prior light history standardisation, participants performed executive (always first), emotional and attentional tasks (pseudo-randomly 2nd or 3rd, blue arrow). As the attentional task included fewer light conditions, it is not considered in the present manuscript (see Materials and methods for more details). (**B**) Spectral power distribution of light exposures. Monochromatic orange: 0.16 mel EDI lux; Polychromatic, blue-enriched light (6500 K); LOW, MID, HIGH: 37, 92, 190 mel EDI lux. For the present analyses, we discarded colour differences between the light conditions and only considered illuminance as indexed by mel EDI lux, constituting a limitation of our study. See *Supplementary file 1b* for full details). (**C, D**) Tasks procedures. Time is reported in seconds relative to session onset; participants were pseudo-randomly exposed to the 4 light conditions. (**C**) Executive task: alternation of letter detection blocks (0-back) and working memory blocks (2-back). (**D**) Emotional task: lure gender discrimination of vocalisations (50% angry (red), 50% neutral (white).

unspecific and widespread increase in BOLD signal surrounding the hypothalamus subparts and that the effect of light was most prominent over the posterior and inferior-anterior subparts.

## Opposite dynamics between the posterior and inferior/anterior hypothalamus at higher illuminance

This prompted us to assess the activity of the hypothalamus subparts under each illuminance to detail the different regional activity dynamics across the hypothalamus. The statistical analyses confirmed that the activity dynamics across illuminance levels differed between the five subparts during the executive and the emotional tasks (GLMM; subparts-by-illuminance interaction; p=0.041) tasks (*Figure 2H–K*; *Table 2*). Post hoc contrasts first considered the impact of the changes in illuminance within each

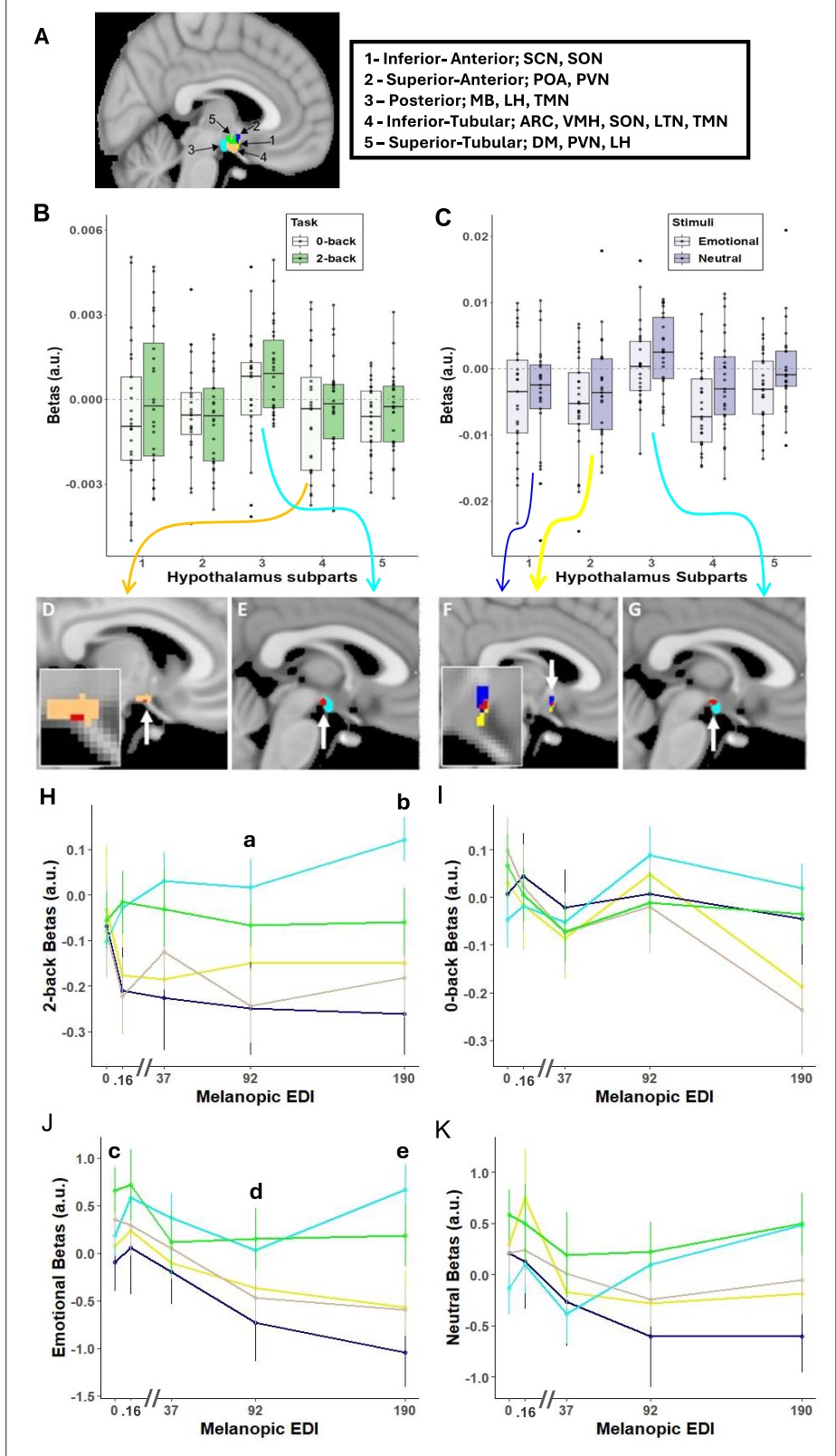

**Figure 2.** Illuminance impact on the hypothalamus subparts. (**A**) Segmentation of the hypothalamus in five subparts in a representative participant. The nuclei encompassed by the different subparts are indicated in the right inset – according to ***Billot et al., 2020***. ARC: arcuate nucleus; DMH; dorsomedial nucleus; LH lateral hypothalamus; LTN: lateral tubular nucleus; MB: mamillary body; POA: preoptic area; PVN: paraventricular

*Figure 2 continued on next page*

*Figure 2 continued*

nucleus; PNH: posterior nucleus of the hypothalamus; SCN: suprachiasmatic nucleus; SON: supraoptic nucleus; TMN: tuberomammillary nucleus; VMN: ventromedial nucleus. (**B, C**) Estimates (beta; arbitrary unit – a.u.) of the collective impact of illuminance variation on the activity of each hypothalamus subpart (Refer to *Table 1* full statistics). (**B**) Executive task: significant main effect of hypothalamus subparts (p=0.002), no significant main of task type (p=0.4) or subpart-by-task-type interaction (p=0.61). (**C**) Emotional task: significant main effect of hypothalamus subparts (p<0.0001), and of stimulus type (p=0.048) or subpart-by-stimulus-type interaction (p=0.74). (**D–G**) Whole brain analyses of the collective impact of the variations in illuminance over the hypothalamus area - for illustration. A local positive peak (red; $p_{uncorrected}$ <0.001) was detected over the posterior hypothalamus subpart (light blue) in executive (**E**) and emotional (**G**). A local negative peak (red; $p_{uncorrected}$ <0.001) was detected over the inferior-tubular hypothalamus subparts (light orange) during the executive task (**D**), while local negative peak (red; $p_{uncorrected}$ <0.001) was detected over the inferior-anterior (yellow) and superior-anterior (blue) hypothalamus subparts during the emotional task (**F**) – insets correspond to enlargements over the hypothalamus area. Arrows from panels B and C arise from and are colour coded according to the hypothalamus subpart that is displayed in panels D to G. These results indicate that our finding does not arise from a nearby 'leaking' activation/deactivation. (**H–K**) Estimates of the impact of each illuminance on the activity of the hypothalamus subparts. (Refer for *Table 2* and *Supplementary file 1c–f* for full statistics) Activity dynamics across illuminance for each subpart (colour code as in A). Results are displayed per task or stimulus type although no interactions with task or stimulus type were detected. Significant illuminance-by-hypothalamus-subpart interactions were detected for (**H, I**) the executive task (p=0.041) and (**J, K**) the emotional task (p=0.041). Small letter indicate significant difference (p<0.05) between the following subparts at illuminance: a. 92 mel EDI lux: posterior vs. superior-anterior & inferior-tubular; b. 190 mel EDI lux: posterior vs. inferior-anterior, superior-anterior and inferior-tubular; c. 0 mel EDI lux: posterior vs. superior-tubular; d. 92 mel EDI lux: posterior vs. superior-anterior; superior-anterior vs. superior-tubular; e. 190 mel EDI lux: posterior vs. inferior-anterior, superior-anterior and inferior-tubular; superior-tubular vs. superior-anterior, inferior-tubular and inferior-anterior. Means +- standard deviations are plotted.

subpart (*Supplementary file 1c and d*). The activity of the posterior hypothalamus subpart significantly (p<0.05) increased under the highest illuminance (190 mel EDI) compared with darkness for both tasks and with the lower illuminances (37 and 92 mel EDI lux) for the emotional task. In contrast, for both tasks, the activity in the inferior-anterior and inferior-tubular hypothalamus subparts significantly (p<0.05) decreased under the highest illuminance compared with darkness, and with lower illuminances for the emotional task. Finally, the activity of the superior anterior hypothalamus subpart decreased under higher illuminance during the emotional but not the executive task, while the activity of the fifth hypothalamus subpart, the superior tubular subpart, was not significantly affected by illuminance changes in either task.

Post hoc analyses also yielded several significant differences between hypothalamus subparts (p<0.05; *Table 2*; *Supplementary file 1e and f*). For both tasks, the activity of the posterior hypothalamus subpart was consistently significantly higher than the activity inferior-tubular subpart under the highest illuminances (92 and 190 mel EDI lux). For the executive task, the activity of the posterior hypothalamus subpart was also significantly higher than the superior-anterior subpart under the highest illuminances (92 and 190 mel EDI lux). For the emotional task, the activity of the posterior hypothalamus subpart was also significantly higher than the superior-anterior subpart under the highest illuminances (190 mel EDI lux), while the activity superior-tubular hypothalamus subpart was significantly higher than the activity of the inferior-tubular, inferior-anterior and superior-anterior hypothalamus subparts (92 and/or190 mel EDI lux). The overall picture arising from these comparisons is that higher illuminance increased the activity of the posterior and superior hypothalamus subparts while it decreased the activity of the inferior and anterior hypothalamus subparts.

## Performance to the executive task is improved by light and related to the activity of the posterior hypothalamus

Following these analyses, we explored whether the changes in activity across illuminances were related to cognitive performance. We first considered the more difficult (2-back) subtask of the executive task as it requires higher cognitive functions (see Materials and methods for a full rationale; *Collette et al., 2005*). The analysis revealed that accuracy to the executive task was high in all participants, but accuracy to the more difficult subtask (2-back) improved with increasing illuminance (GLMM; main effect of illuminance; *F*=2.72; p=0.034; Partial R2 215=0.1; *Figure 3A*), controlling for

**Table 1.** Differences between hypothalamus subparts in the collective impact of the variation in illuminance on their activity.

**Executive task**

| Main GLMM | | | | Pairwise comparisons | | | |
|---|---|---|---|---|---|---|---|
| Effect | F value (df) | p value[*] | Partial $R^2$ | Contrast[†] | t-value | $P_{uncorrected}$ | $P_{corrected}$ |
| | | | | 1 vs 2 | –0.30 | 0.76 | 0.99 |
| | | | | 1 vs 3 | –3.48 | 0.0006 | 0.0056 |
| Hypothalamus subparts | 4.36 (4,225) | 0.002 | 0.08 | 1 vs 4 | <0.01 | 0.99 | 1 |
| Task | 0.74 (1,25) | 0.4 | | 1 vs 5 | –0.57 | 0.57 | 0.98 |
| | | | | 2 vs 3 | –3.17 | 0.0017 | 0.015 |
| | | | | 2 vs 4 | 0.31 | 0.76 | 0.99 |
| Hypothalamus subparts x task type | 0.68 (4,225) | 0.61 | | 2 vs 5 | –0.27 | 0.79 | 0.99 |
| Age | 0.33 (1,22) | 0.57 | | 3 vs 4 | 3.48 | 0.0006 | 0.0055 |
| BMI | 0.59 (1,22) | 0.45 | | 3 vs 5 | 2.54 | 0.0041 | 0.033 |
| Sex | 0.01 (1,22) | 0.91 | | 4 vs 5 | –0.5 | 0.57 | 0.98 |

**Emotional task**

| Main GLMM | | | | Pairwise comparisons | | | |
|---|---|---|---|---|---|---|---|
| Effect | F Value | p value[*] | Partial $R^2$ | Contrast[†] | t-value | $P_{uncorrected}$ | $P_{corrected}$ |
| | | | | 1 vs 2 | 0.67 | 0.67 | 0.99 |
| | | | | 1 vs 3 | –4.76 | <0.0001 | <0.0001 |
| Hypothalamus subparts | 9.38 (4,194) | <.0001 | 0.22 | 1 vs 4 | 0.00 | 0.99 | 1 |
| Task | 4.33 (1,25) | 0.048 | 0.15 | 1 vs 5 | –1.88 | 0.06 | 0.33 |
| | | | | 2 vs 3 | –5.43 | <0.0001 | 0.0001 |
| | | | | 2 vs 4 | –0.66 | 0.51 | 0.96 |
| Hypothalamus subparts x stimulus type | 0.5 (4,194) | 0.74 | | 2 vs 5 | –2.54 | 0.012 | 0.086 |
| Age | 0.43 (1,22) | 0.52 | | 3 vs 4 | 4.76 | <0.0001 | <0.0001 |
| BMI | 0.05 (1,22) | 0.83 | | 3 vs 5 | 2.85 | 0.0048 | 0.038 |
| Sex | 1.47 (1,22) | 0.24 | | 4 vs 5 | –1.88 | 0.061 | 0.32 |

Outputs of the generalized linear mixed model (GLMM) with subject as the random factor (intercept and slope), and task and subpart as repeated measures (ar(1) autocorrelation).

[*]The corrected p-value for multiple comparisons over 2 tests is p < 0.025.

[†]Refer to **Figure 2A** for correspondence of subpart number.

age, sex, and BMI. Critically, the analysis also showed that performance under each illuminance was significantly related to the activity of the posterior hypothalamus subpart (GLMM; main effect of posterior subpart activity; $F=9.43$; $p=0.0027$; Partial R2 219=0.09). Surprisingly, the association was negative (**Figure 3B**), suggesting that the part of variance explained by the hypothalamus subpart is distinct from the impact of light on performance. In contrast, no significant association was found when considering the activity of the other four subparts (GLMM; main effect of posterior subpart activity; $F<0.62$; $p>0.4$; **Figure 3C and D**; **Supplementary file 1g**). We went on and found that the accuracy to the simpler control subtask of the executive tasks (0-back, see Materials and methods) was not associated with the activity of the posterior hypothalamus subpart (GLMM controlling for age, sex and BMI; main effect of subpart activity; $F=0.57$; $p=0.45$; **Figure 3E**), suggesting that the association with performance is specific to the 2-back subtask.

In the last step, we explored the reaction times during the emotional task (accuracy to the lure task is not meaningful). We found that reaction times to the emotional stimuli were not significantly affected by illuminance (GLMM controlling for age, sex and BMI; main effect of illuminance; $F=1.01$;

**Table 2.** Statistical outputs of GLMM testing for differences between the activity of each subpart of the hypothalamus under each illuminance.

**Executive task**

| Main GLMM | | | | Comparisons between subparts per illuminance[†] | | | |
|---|---|---|---|---|---|---|---|
| Effect | F-value (df) | P value | Partial R² | Illuminance* | contrast | t-value | p-value |
| Subpart | 1.4 (4,228) | 0.23 | | 92 | 2 vs 3 | –2.25 | 0.025 |
| | | | | 92 | 3 vs 4 | 2.58 | 0.01 |
| Illuminance | 2.15 (4,1017) | 0.073 | | 190 | 1 vs 3 | –2.80 | 0.0053 |
| Task | 3.24 (1,228) | 0.073 | | 190 | 2 vs 3 | –2.24 | 0.025 |
| | | | | 190 | 3 vs 4 | 3.15 | 0.0017 |
| Subpart x Illuminance | 1.7 (16,1017) | 0.041 | 0.09 | | | | |
| Age | 1.19 (1,22) | 0.29 | | | | | |
| BMI | 0.01 (1,22) | 0.9 | | | | | |
| Sex | 0.38 (1,22) | 0.54 | | | | | |

**Emotional task**

| Main GLMM | | | | Comparisons between subparts per illuminance[†] | | | |
|---|---|---|---|---|---|---|---|
| Effect | F-value (df) | p value | Partial R² | Illuminance* | contrast | t-value | p-value |
| | | | | 0 | 3 vs 5 | –2.05 | 0.04 |
| Subpart | 4.29 (4,229) | 0.0023 | 0.07 | 92 | 2 vs 3 | –2.53 | 0.012 |
| | | | | 92 | 2 vs 5 | –2.96 | 0.0032 |
| | | | | 190 | 1 vs 3 | –3.31 | 0.001 |
| Illuminance | 9.41 (4,1020) | <0.0001 | 0.035 | 190 | 2 vs 3 | –4.75 | <0.0001 |
| | | | | 190 | 1 vs 5 | –2.5 | 0.013 |
| | | | | 190 | 2 vs 5 | –4.04 | <0.0001 |
| Task | 0.13 (1,229) | 0.72 | | 190 | 3 vs 4 | 3.13 | 0.0018 |
| Subpart x Illuminance | 1.7 (16,1020) | 0.041 | 0.026 | 190 | 4 vs 5 | –2.32 | 0.021 |
| Age | 0.59 (1,22) | 0.45 | | | | | |
| BMI | 1.54 (1,22) | 0.23 | | | | | |
| Sex | 0.05 (1,22) | 0.83 | | | | | |

* illuminance in mel EDI lux.

[†] Only significant comparisons are reported in the main text. For the full table, including post hocs comparing light levels within a subpart, refer to **Supplementary file 1c–g**.

p=0.41; *Figure 3F*) and yet, they were significantly associated with the activity of the posterior hypothalamus subpart across each illuminance (GLMM; main effect of subpart activity; F=4.34; p=0.04; *Figure 3G*). The association was positive meaning that reaction times were longer if activity estimates were higher, which could indicate a reinforcement of the emotional response characterised by longer reaction times (*Grandjean et al., 2005*). No such significant association was detected when considering reaction times to the neutral items of the task (GLMM controlling for age, sex and BMI; main effect of illuminance; F=1.5; p=0.21; main effect of subpart activity; F=0.28; p=0.6; *Figure 3H*).

## Discussion

Animal research has established that the biological impact of light illuminance impinges on many subcortical structures, many of which regulate sleep and wakefulness (*Campbell et al., 2023*; *Hattar*

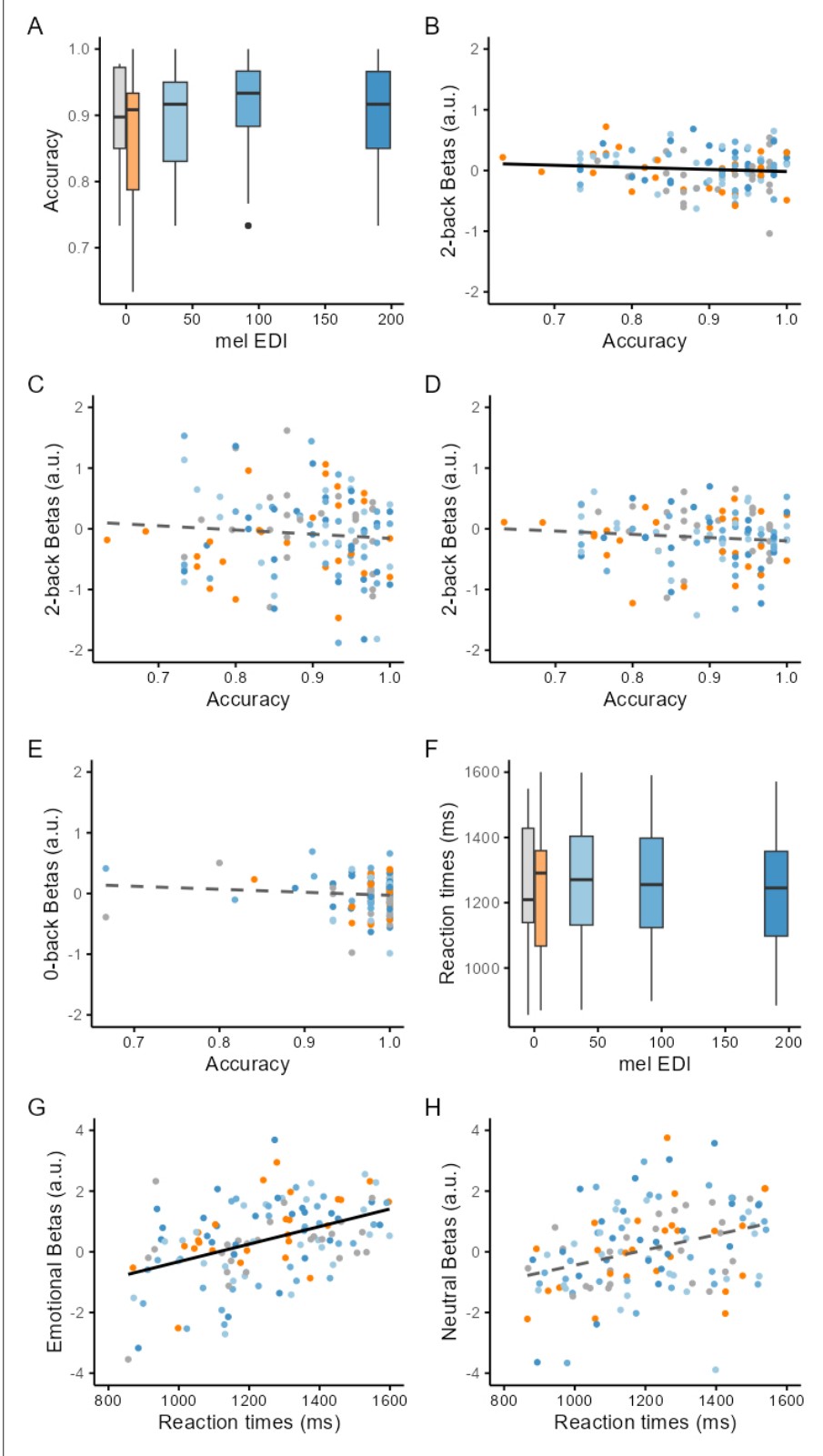

**Figure 3.** Impact of illuminance on performance and relationships with the activity of the posterior hypothalamus subpart. (**A**) Accuracy (percentage of correct responses) to the 2-back increased with increasing illuminance (p=0.034). (**B**) Accuracy to the 2-back task is negatively correlated to the activity of the posterior hypothalamus subpart (p=0.0027). (**C, D**) Accuracy to the 2-back task is not correlated to the activity of the inferior-anterior

*Figure 3 continued on next page*

*Figure 3 continued*

(**C**) and inferior-tubular (**D**) hypothalamus subparts (p>0.4). Association between superior-anterior and superior-tubular subparts are not displayed but were not significant (p>0.6). See ***Supplementary file 1g*** for full details. (**E**) Accuracy to the 0-back task is not correlated to the activity of the posterior hypothalamus subpart (p=0.45). (**F**) Reaction times to the emotional stimuli did not significantly change with increasing illuminance (p=0.41). (**G**) Reaction times to the emotional stimuli are correlated to the activity of the posterior hypothalamus subpart (p=0.04) with higher activity associated to slower reaction times. (**H**) Reaction times to the neutral stimuli are not correlated to the activity of the posterior hypothalamus subpart (p=0.6). Box plots: horizontal line = median; box = higher/lower quartiles: vertical line = maximum/minimum. Regression plots: solid and dashed lines correspond to the significant and not significant linear regression lines, respectively.

***et al., 2006***; ***Scammell et al., 2017***; ***Do, 2019***). How these findings translate to human beings is not established. Here, we took advantage of the relatively high resolution and signal-to-noise ratio of UHF 7T fMRI to determine how illuminance affects the activity of the hypothalamus as, based on animal research, it receives the densest projections from ipRGCs (***Hattar et al., 2006***; ***Do, 2019***). We find that the activity of the posterior part of the hypothalamus increases with increasing illuminance while, in contrast, the inferior and anterior hypothalamus show a seemingly opposite pattern and see their activity decrease under higher illuminance. The pattern of light-induced changes was consistent across an executive and an emotional task which consisted of a block and an event-related fMRI design, respectively. This suggests that a robust anterior-posterior gradient of activity modulation by illuminance is present in the hypothalamus across cognitive domains. Importantly, performance to the complex cognitive task was improved under higher illuminance and was correlated to the activity of the posterior part of the hypothalamus under the different illuminance, though negatively. The results demonstrate that the human hypothalamus does not respond uniformly to variations in illuminance while engaged in a cognitive challenge and suggest that the posterior part of the hypothalamus may be key in mediating the stimulating impact of light on cognition.

The different nuclei of the hypothalamus do not have clear contrast boundaries based on MRI signals (***Billot et al., 2020***). As a result, achieving nucleus resolution over the human hypothalamus even using UFH MRI remains out of reach (***Sharifpour et al., 2022***). Therefore, we cannot assign the effects we report to a specific nucleus. We can only speculate and present a selection of plausible scenarios that would need to be tested. The posterior part of the hypothalamus – delineated in each participant based on an automatic reproducible procedure (***Billot et al., 2020***) – encompasses the MB as well as parts of the LH and the TMN. All these nuclei could participate to the increased BOLD signal we detect under higher illuminance. The LH and TMN, respectively, produce orexin and histamine, which are both known to promote wakefulness, while animal histology reports direct projection of the ipRGCs to the LH (***Hattar et al., 2006***; ***Scammell et al., 2017***; ***Do, 2019***). Orexin is a good candidate to constitute the circadian signal that promotes wakefulness and to counter the progressive increase in sleep needs with prolonged wakefulness (***Zeitzer, 2013***). Our data may therefore be compatible with an increase in orexin release by the LH with increasing illuminance. In line with this assumption, chemoactivation of ipRGCs lead to increase c-fos production, a marker of cellular activation, over several nuclei of the hypothalamus, including the lateral hypothalamus (***Milosavljevic et al., 2016***). If the initial effect of light we observe over the posterior part of the hypothalamus was maintained over a longer period of exposure, this would stimulate cognition and maintain or increase alertness (***Campbell et al., 2023***) and may also be part of the mechanisms through which daytime light increases the amplitude in circadian variations of several physiological features (***Bano-Otalora et al., 2021***; ***Dijk et al., 2012***). It could then also be part of the mechanisms through which evening light may disturb subsequent sleep (***Chellappa et al., 2013***) when illuminance is higher than the recommended maximum of 10 mel EDI lux for evening light (***Brown et al., 2022***). If the TMN was the hypothalamus nucleus underlying the regional increase in the BOLD signal we report, it could confer a role to histamine in mediating the stimulating impact of light. Of interest, the TMN receives orexin signal from the LH (***Scammell et al., 2019***). Alternatively, our findings may suggest a role for the MBs in mediating the impact of light on ongoing cognition, potentially influenced through its innervation by the TMN (***Vann, 2010***).

Previous research indicated that increasing illuminance reduced the activity of the anterior part of the hypothalamus encompassing the SCN, either following the exposure to light (***Perrin et al., 2004***)

or during the exposure (*Schoonderwoerd et al., 2022*). We extend this finding by showing that the significant decrease in activity extends beyond the inferior anterior hypothalamus and therefore much beyond the SCN. Chemoactivation of ipRGCs in rodents led to an increase activity of the SCN, over the inferior anterior hypothalamus, but had no impact on the activity of the VLPO, over the superior anterior hypothalamus (*Milosavljevic et al., 2016*). How our findings fit with these fine-grained observations and whether there are species-specific differences in the responses to light over the different part of the hypothalamus remains to be established. The inferior-tubular and inferior-anterior subparts of the hypothalamus – that we isolated also based on an automatic reproducible procedure (*Billot et al., 2020*) – encompass several nuclei such as notably the SCN, SON, ventromedial nucleus of the hypothalamus, arcuate nucleus and part of the TMN. Again, all these nuclei may be involved in the reduction in BOLD signal we observe at higher illuminance. In terms of chemical communication, these changes in activity could be the results of an inhibitory signal from a subclass of ipRGCs, potentially through the release aminobutyric acid (GABA), as a rodent study found that a subset of ipRGCs release GABA at brain targets including the SCN (and intergeniculate leaflet and ventral lateral geniculate nucleus), leading to a reduction in the ability of light to affect pupil size and circadian photoentrainment (*Sonoda et al., 2020*). Whatever the signalling of ipRGC, our finding over the anterior hypothalamus could correspond to a modification of GABA signalling of the SCN which has been reported to have excitatory properties, such that the BOLD signal changes we report may correspond to a reduction in excitation arising in part from the SCN (*Albers et al., 2017*). Likewise, the SCN is also producing other neuropeptides that could affect its downstream targets. As the inferior-tubular subpart of the hypothalamus also includes part of the TMN it may be the TMN and its GABA production that is decreased by higher illuminance (*Scammell et al., 2019*). The decrease in BOLD signal with increasing illuminance we report could therefore arguably reflect a decreased inhibitory signal arising from the anterior and inferior nuclei of the hypothalamus.

Importantly, none of the scenarios we elaborated on are mutually exclusive and we may have overlooked the potential implication of several nuclei as well as the cellular diversity of the nuclei of the hypothalamus (*Adamantidis et al., 2019*; *Scammell et al., 2017*). We further note that the anterior-superior hypothalamus subpart of the hypothalamus encompassing the VLPO and PON sees its activity decreasing under higher illuminance during the emotional task, similarly to the anterior-inferior and inferior-tubular areas. Likewise, similar to the posterior subpart, the activity of the superior-tubular hypothalamus subpart may be increased under higher illuminance during the emotional task. Whether this represents a task-specific effect arising, for instance, from differences in the salience of the auditory stimulus, remains to be determined.

Subcortical structures, and particularly those receiving direct retinal projections, including those of the hypothalamus, are likely to receive light illuminance signal first, before passing on the light modulation to the cortical regions involved in the ongoing cognitive process (*Campbell et al., 2023*). A critical aspect of our results is that the performance to the 2-back (executive) subtask was significantly increased when exposed to higher illuminance light. The extent of this increase was limited, likely because performance was overall high at all illuminances, but was not detected for the simpler detection letter subtask (0-back). The result contrasts with many previous 3T MRI investigations on the biological effects of light on human brain function which did not report behavioural changes induced by repeated short exposures to light (e.g. *Vandewalle et al., 2007a*; *Vandewalle et al., 2007b*; *Vandewalle et al., 2011a* but see *Daneault et al., 2018*; *Vandewalle et al., 2006*). Our 7T MRI study, which includes a sample size larger than many of these previous studies, supports that BOLD fMRI is sensitive in detecting subtle impacts of light on the brain and that these detected changes can arguably contribute to the behavioural changes others reported using longer light exposure and other approaches (e.g. *Cajochen et al., 2011*; *Lockley et al., 2006* but see *Smolders et al., 2018*).

Importantly, we find that activity of the posterior hypothalamus subpart is negatively related to the performance to the executive task, making it unlikely that it mediates directly the positive impact of light on performance. The activity of the posterior hypothalamus was, however, associated with an increased behavioural response to emotional stimuli. The association between behaviour and the posterior hypothalamus is therefore likely to be complex and may depend on the context, with for instance different nuclei or neuronal populations contributing in some instances but not in others (*Adamantidis et al., 2019*; *Scammell et al., 2019*). It is likely also to work jointly with the decreased activity of the anterior/inferior hypothalamus we detected as well as with other non-hypothalamus

subcortical structures regulating wakefulness to influence behaviour, which intrinsically primarily depends on cortical activity.

Future research may consider data-driven analyses of hypothalamus voxels time series as an alternative to the parcellation approach we adopted here. This may refine the delineation of the subparts of the hypothalamus undergoing decreased or increased activity with increasing illuminance. Future research should also assess the impact of light on other subcortical structures and on the entire subcortical network to determine how illuminance modifies their crosstalk as well as their interaction with the cortex, to eventually lead to behavioural impacts. These analyses could for instance address whether the regional changes in activity of the hypothalamus we find are upstream of the repeatedly reported impact of light illuminance on the activity of the pulvinar in the thalamus (*Paparella et al., 2023*; *Vandewalle et al., 2006*). Although it does not receive direct dense input from ipRGCs, it is likely to indirectly mediate the biological impact of light on ongoing cognitive activity (*Paparella et al., 2023*).

We based our rationale and part of our interpretations on ipRGC projections, which have been demonstrated in rodents to channel the NIF biological impact of light and incorporate the inputs from rods and cones with their intrinsic photosensitivity into a light signal that can impact the brain (*Güler et al., 2008*; *Do, 2019*). Given the polychromatic nature of the light we used, classical photoreceptors and their projections to visual brain areas are, however, very likely to have directly or indirectly contributed to the modulation by light of the regional activity of the hypothalamus. Furthermore, we cannot exclude that colour and/or spectral differences between the orange and three blue-enriched light conditions may have contributed to our findings. Research in rodent models demonstrated that variation in the spectral composition of light was perceived by the suprachiasmatic nucleus to set circadian timing (*Walmsley et al., 2015*). No such demonstration has, however, been reported yet for the acute impact of light on alertness, attention, cognition or affective state. Future human studies could isolate the contribution of each photoreceptor class to the impact of light on cognitive brain functions by manipulating prior light history (*Chellappa et al., 2014*) or through the use of silent substitutions between metameric light exposures (*Viénot et al., 2012*).

All these knowledge gaps are important to address because acting on light stands as a promising means to reduce high sleepiness and improve cognitive deficits during wakefulness as well as to facilitate sleep in the few hours preceding bedtime (*Brown et al., 2022*; *Didikoglu et al., 2023*; *Gooley et al., 2011*; *Münch et al., 2016*; *Riemersma-van der Lek et al., 2008*; *Scheuermaier et al., 2018*; *Wirz-Justice et al., 2021*). Light therapy is also a validated means to improve mood and treat mood disorders (*Glickman et al., 2006*; *Lam et al., 2016*). Light administration can also be considered a simple means to disturb the brain circuitry regulating sleep and wakefulness such that it can provide insights about novel means to improve their quality. For instance, if orexin and histamine were part of the mechanism through which natural light affects brain functions, their administration may be the most ecological and/or natural means to affect alertness and cognition. Likewise, as both orexin and histamine are targets for the treatment of brain disorders, our findings could suggest that light may constitute a non-pharmacological complementary intervention to compounds that are being developed to treat arousal, sleep, or cognitive dysfunction in brain disorders (*Ma et al., 2018*). It remains, however, premature in our view to base recommendations on the therapeutic use of light based on the MRI findings gathered to date. Targeted lighting for interventions or for precise interference of subcortical circuits will require a full understanding of how light affects the brain, particularly at the subcortical level. Our findings represent an important step towards this goal, at the level of the hypothalamus.

## Materials and methods

The data used in this paper arise from a large study that is leading to several publications and part of the methods have been published previously (*Beckers et al., 2024*; *Campbell et al., 2024a*; *Paparella et al., 2023*). The protocol was approved by the Ethics Committee of the Faculty of Medicine at the University of Liège (ref 2020/11). Participants gave their written informed consent to take part in the study and received monetary compensation for their participation.

## Participants

Thirty healthy young adults (19 women; 24.3±2.9 y; *Supplementary file 1*) were included in the analyses. Exclusion criteria were assessed through questionnaires and a semi-structured interview: history of psychiatric and neurological disorders, sleep disorders, use of psychoactive drugs or addiction; history of ophthalmic disorders or auditory impairments; colour blindness; night shift work during the last year or recent trans-meridian travel during the last 2 months; excessive caffeine (>4 caffeine units/day) or alcohol consumption (>14 alcohol units/week); medication affecting the central nervous system; smoking; pregnancy or breast-feeding (women); counter indication for MRI-scanning. All participants had to score <18 on the 21-item Beck Anxiety Inventory (up to mild anxiety; *Beck et al., 1988a*), and <14 on the Beck Depression Inventory-II (up to mild depression; *Beck et al., 1988b*), <12 on the Epworth Sleepiness Scale (*Johns, 1993*), and <8 on the Pittsburgh Sleep Quality Index (*Buysse et al., 1989*). Questionnaires further assessed chronotype with the Horne-Östberg questionnaire (*Horne and Ostberg, 1976*) and seasonality with the Seasonal Pattern Assessment Questionnaire (*Rosenthal and Bradt, 1984*), but the latter two questionnaires were not used for the inclusion of the participants.

For each task, four datasets were missing or had corrupt data such that 26 participants were included in the analyses of each task (23 participants had valid datasets for both tasks). For the emotional task, two participants' data failed the MRI quality control (QC) check, and the other two participants were excluded as they did not complete the entire task. For the executive task, four of the participants' data failed the MRI QC check. *Supplementary file 1b* summarises participants' characteristics respective to each task.

## Overall protocol

Participants completed an MRI session at least one week before the experiment during which structural images of the brain were acquired and which served as habituation to the experimental conditions. Participants maintained a loose sleep-wake schedule (±1 hr from the habitual sleep/wake-up time) during the 7 days preceding the fMRI experiment to warrant similar circadian entrainment across participants and avoid excessive sleep loss while maintaining realistic real-life conditions (verified using sleep diaries and wrist actigraphy - AX3 accelerometer, Axivity, United Kingdom). Volunteers were requested to refrain from all caffeine and alcohol-containing beverages, and extreme physical activity for 3 days before participating in the fMRI acquisitions. Data acquisitions took place in Liège, Belgium, between December 2020 and May 2023.

Participants arrived at the laboratory 1.5–2 hr after habitual wake time for the fMRI scan. They were first exposed for 5 min to a bright polychromatic white light (1000 lux) and then maintained in dim light (<10 lux) for 45 min to standardise the participant's recent light history. During this period, participants were given instructions about the fMRI cognitive tasks and completed practice tasks on a luminance-controlled laptop (<10 lux). The fMRI session consisted of participants completing three auditory cognitive tasks while alternatively maintained in darkness or exposed to light: an executive task (25 min), an emotional task (20 min) and an attentional task (15 min) [*Figure 1A*]. The executive task was always completed first, as it was the most demanding task. The order of the following two tasks was counterbalanced. Because it included only three light conditions (see below) instead of five for the other two tasks, the attentional task was not included in the present analyses. An eye-tracking system (EyeLink 1000Plus, SR Research, Ottawa, Canada) was monitored for proper eye opening during all data acquisitions.

## Light exposure

An 8 m long MRI-compatible optic fibre (1-inch diameter, Setra Systems, MA, USA) transmitted light from a light box (SugarCUBE, Ushio America, CA, USA) to the dual end of the fibre which was attached to a stand fitted at the back of the MRI coil that allowed reproducible fixation and orientation of the optic fibre ends. The dual branches illuminated the inner walls of the head coil to ensure relatively uniform and indirect illumination of participants' eyes. A filter wheel (Spectral Products, AB300, NM, USA) and optical fibre filters (monochromatic narrowband orange filter - 589mn; full width at half maximum: 10 nm - or a UV highpass filter - 433–1650 nm) were used to create the light conditions needed for the experiment (see *Figure 1B* and *Supplementary file 1b* for in-detail light characteristics).

Illuminance and spectra could not be directly measured within the MRI scanner due to the ferromagnetic nature of measurement systems. The coil of the MRI and the light stand, together with the lighting system were therefore placed outside of the MR room to reproduce the experimental conditions of the in a completely dark room. A sensor was placed 2 cm away from the mirror of the coil that is mounted at eye level, that is where the eye of the first author of the paper would be positioned, to measure illuminance and spectra. The procedure was repeated four times for illuminance and twice for spectra and measurements were averaged. This procedure does not take into account inter-individual variation in head size and orbit shape such that the reported illuminance levels may have varied slightly across subjects. The relative differences between illuminance are, however, very unlikely to vary substantially across participants such that statistics consisting of tests for the impact of relative differences in illuminance were not affected. The detailed values reported in *Supplementary file 1b* were computed combining spectra and illuminance using the excel calculator associated with a published work (*Lucas et al., 2014*).

Blue-enriched light illuminances were set according to the technical characteristics of the light source and to keep the overall photon flux similar to prior 3T MRI studies of our team (between ~$10^{12}$ and $10^{14}$ ph/cm²/s *Vandewalle et al., 2010*; *Vandewalle et al., 2011b*). The orange light was introduced as a control visual stimulation for potential secondary whole-brain analyses. For the present region of interest analyses, we discarded colour differences between the light conditions and only considered illuminance as indexed by mel EDI lux. This constitutes a limitation of our study as it does not allow attributing the findings to a particular photoreceptor class.

For the executive and emotional task, the light conditions consisted of three different illuminance of a white, blue-enriched polychromatic LED light (37, 92, 190 mel EDI lux; 6500 K) and one illuminance level of monochromatic orange light (.16 mel EDI lux; 590 nm full width at half maximum - FWHM: 10 nm). For the present analyses, we discarded colour differences between the light conditions and only considered illuminance as indexed by mel EDI lux, constituting a limitation of our study. In the executive task, participants were exposed to 30s to 70s (median 30 s) of light blocks separated by 10 s of darkness (<0.1 lux) and the light blocks were repeated 11 times for each light condition. For the emotional task, participants were exposed to 30–40 s (median 35 s) light blocks separated by 20 s of darkness (<0.1 lux) and the light blocks were repeated five times for each light condition.

The attentional task only included a single illuminance level of the blue-enriched polychromatic LED light (92 mel EDI lux) and one illuminance level of the monochromatic orange light (.16 mel EDI lux), otherwise, the task would have been too long (>30 min). Participants were exposed to 30 s of light blocks separated by 10 s of darkness (<0.1 lux). The light blocks were repeated 7 times for each light condition. As mentioned above it is not considered for the present analyses.

## Cognitive tasks

Prior work of our team showed that the n-back task and emotional task included in the present protocol were successful probes to demonstrate that light illuminance modulates cognitive activity, including within subcortical structures (although resolution did not allow precise isolation of nuclei or subparts e.g. *Vandewalle et al., 2010*; *Vandewalle et al., 2007b*). When taking the step of ultra-high-field imaging, we therefore opted for these tasks as our goal was to show that illuminance affects brain activity across cognitive domains while not testing for task-specific aspects of these domains.

## Auditory cognitive tasks

The tasks were programmed with Opensesame (3.2.8 *Mathôt et al., 2012*). Participants heard the auditory stimuli through MR-compatible earbuds (Sensimetrics, Malden, MA). Before starting the tasks, to ensure optimal auditory perception of task stimuli, participants set the volume through a volume check procedure. Participants used an MRI-compatible keypad to respond to task items (Current Designs, Philadelphia, PA), which was placed in the participant's dominant hand. The tasks were separated by about 5 min in near darkness, to recalibrate the eye tracking system and to clarify instructions about the next task to the participant.

## Executive task

The task consisted of an auditory variant of the n-back task (*Collette et al., 2005*) with a working memory 2-back task and a control letter detection 0-back task. Participants were either asked to

detect whether the current item was identical to the letter presented 2-items earlier (2-back) or whether the current item consisted of the letter 'K' (0-back) or using the keypad (one button for 'yes', one button for 'no'). A block design was used for this task in which each block included 15 items and lasted 30 s. Task blocks were separated by 10–20 s rest periods and were preceded by an auditory instruction (500ms) indicating the type of task to be completed. Task levels were pseudo-randomised across the four light conditions with three blocks of 0-back and four blocks of 2-back per light condition (see *Figure 1C*).

## Emotional task

The task consisted of gender discrimination of auditory vocalisations that were either pronounced with emotional or neutral prosody (*Grandjean et al., 2005*). Participants were asked to use the keypad to indicate what they believed the gender of the person pronouncing each token was. The gender classification was a lure task ensuring participants paid attention to the auditory stimulation. The purpose of the task was to trigger an emotional response as participants were not told that part of the stimuli was pronounced with angry prosody. The 240 auditory stimuli were pronounced by professional actors (50% women) and consisted of three meaningless words ('goster', 'niuvenci', 'figotleich'). The stimuli were expressed in either an angry or neutral prosody, which has been validated by behavioural assessments (*Banse and Scherer, 1996*) and in previous experiments (*Grandjean et al., 2005*; *Sander et al., 2005*; *Vandewalle et al., 2010*). The stimuli were also matched for the duration (750ms) and mean acoustic energy to avoid loudness effects. During each 30–40 s light block, four angry prosody stimuli and four neutral prosody stimuli were presented in a pseudorandom order and delivered every 3–5 s. A total of 160 distinct voice stimuli (50% angry; 50% neutral) were distributed across the four light conditions. The darkness period separating each light block contained two angry and two neutral stimuli. A total of 80 distinct voice stimuli (50% angry; 50% neutral) were distributed across the darkness periods [see *Figure 1D*].

## Data acquisition

The MRI data were acquired in a 7T MAGNETOM Terra MR scanner (Siemens Healthineers, Erlangen, Germany) with a 32-channel receive and 1-channel transmit head coil (Nova Medical, Wilmington, MA, USA). Dielectric pads (Multiwave Imaging, Marseille, France) were placed between the subject's head and receiver coil to homogenise the magnetic field of Radio Frequency (RF) pulses.

Multislice T2\*-weighted fMRI images were obtained with a multi-band Gradient-Recalled Echo - Echo-Planar Imaging (GRE-EPI) sequence using axial slice orientation (TR = 2340ms, TE = 24ms, FA = 90°, no interslice gap, in-plane FoV = 224 mm × 224 mm, matrix size = 160 × 160×86, voxel size = 1.4 × 1.4×1.4 mm$^3$). To avoid saturation effects, the first three scans were discarded. To correct for physiological noise in the fMRI data the participants' pulse and respiration movements were recorded using a pulse oximeter and a breathing belt (Siemens Healthineers, Erlangen, Germany). Following the fMRI acquisition a 2D GRE field mapping sequence to assess B0 magnetic field inhomogeneity with the following parameters: TR = 5.2ms, TEs = 2.26ms and 3.28ms, FA = 15°, bandwidth = 737 Hz/pixel, matrix size = 96 × 128, 96 axial slices, voxel size = (2x2 × 2) mm$^3$, acquisition time = 1:38 min, was applied.

For the anatomical image, a high-resolution T1-weighted image was acquired using a Magnetization-Prepared with 2 RApid Gradient Echoes (MP2RAGE) sequence: TR = 4300ms, TE = 1.98ms, FA = 5°/6°, TI = 940ms/2830ms, bandwidth = 240 Hz, matrix size = 256 × 256, 224 axial slices, acceleration factor = 3, voxel size = (0.75x0.75 × 0.75) mm$^3$.

## Data processing

For the MP2RAGE images, the background noise was removed using an extension (extension: https://github.com/benoitberanger/mp2rage; *Béranger and Papadopoulos Orfanos, 2019*) of Statistical Parametric Mapping 12 (SPM12; https://www.fil.ion.ucl.ac.uk/spm/software/spm12/) under Matlab R2019 (MathWorks, Natick, Massachusetts) (*O'Brien et al., 2014*). Then the images were reoriented using the 'spm_auto_reorient' function (https://github.com/CyclotronResearchCentre/spm_auto_reorient; *Cyclotron Research Centre, 2015*) and corrected for intensity non-uniformity using the bias correction method implemented in the SPM12 'unified segmentation' tool (*Ashburner and Friston, 2005*). To ensure optimal co-registration, brain extraction was done using SynthStrip (*Hoopes et al.,*

*2022*) in Freesurfer (http://surfer.nmr.mgh.harvard.edu/). The brain-extracted T1-images were used to create a T1-weighted group template using Advanced Normalization Tools (ANTs, http://stnava. github.io/ANTs/) prior to normalisation to the Montreal Neurological Institute (MNI) space using ANTs (1 mm³ voxel; MNI 152 template). The hypothalamus of each participant was segmented within 1 mm³ MNI 152 template into five subparts - inferior anterior, superior anterior, inferior tubular, superior tubular, posterior (*Figure 2A*) using an automatic computational approach (*Billot et al., 2020*).

For the EPI images, auto reorientation was applied on the images first. Then, voxel-displacement maps were computed from the phase and magnitude images associated with B0 map acquisition (taken right after the task), using the SPM fieldmap toolbox. To correct for head motion and static and dynamic susceptibility-induced variance, the 'Realign and Unwarp' of SPM12 was then applied to the EPI images. The realigned and distortion-corrected EPI images then underwent brain extraction using the SynthStrip and then the final images were smoothed with a Gaussian kernel characterised by a FWHM = 3 mm. The first level analyses were performed in the native space to prevent any possible error that may be caused by co-registration.

## Statistical analyses

The whole-brain univariate analyses consisted of a general linear model (GLM) computed with SPM12. For the executive task, task blocks and light blocks were modelled as block functions. For the emotional task, the auditory stimuli were modelled as stick functions. For both tasks, a high-pass filter with a 256 s cut-off was applied to remove low-frequency drifts. For both tasks, stick or block functions were convolved with the canonical hemodynamic response function. Movement and physiological parameters (cardiac and respiration), which were computed with the PhysIO Toolbox (Translational Neuromodeling Unit, ETH Zurich, Switzerland), were included as covariates of no interest (*Kasper et al., 2017*).

Two separate analyses were completed. In the main analyses, we sought to test whether brain responses during the tasks were modulated by overall changes in illuminance level. The regressors of task blocks or events were accompanied by a single parametric modulation regressor corresponding to the light melanopic illuminance level (0.16, 37, 92, 190 mel EDI). The contrasts of interest consisted of the main effects of the parametric modulation. In the subsequent post hoc analysis, we estimated the responses to the stimuli under each light condition. Separate regressors modelled each task's block or event type under each light condition (0, 0.16, 37, 92, 190 mel EDI). The contrasts of interest consisted of the main effects of each regressor.

The output masks of the segmentation procedure we used to extract regression betas associated with each of the hypothalamus subparts using the REX Toolbox (https://web.mit.edu/swg/software. htm; *Duff et al., 2007*). Betas were averaged (mean) within each subpart and then across the homologous subparts of each hemisphere. In the main analyses this yielded 1 activity estimate per stimulus type and per hypothalamus subpart (i.e. 10 per individual), while in the subsequent analyses, we obtained 5 activity estimates per stimulus type and per subpart (50 per individual).

For visualisation of whole-brain results over the entire sample, all statistical maps obtained from the first level analysis were first transferred to the group template space and then the MNI space (1x1 × 1 mm³ image resolution). All the registration steps were performed with ANTs. The visualisation was focused on the hypothalamus regions to assess whether increasing illuminance resulted in a local increase and/ or decrease of beta estimates within the hypothalamus or whether beta estimates were mainly influenced by a relatively unspecific and widespread increase in BOLD signal surrounding the hypothalamus.

Statistical analyses of the activity of the hypothalamus subparts were performed in SAS 9.4 (SAS Institute, NC, USA). Analyses consisted of Generalised Linear Mixed Models (GLMM) with the subject as a random factor (intercept and slope) and were adjusted for the dependent variable distribution. As the main statistical analysis was completed for each task, the significance threshold was corrected for multiple comparisons and was set at $p < 0.025$. Direct post hoc of the main analyses were corrected for multiple comparisons using a Tukey adjustment. The subsequent more detailed analyses were considered as post hoc that were not corrected for multiple comparisons ($p < 0.05$). To detect outlier values within the data sets, Cook's distance >1 was used for exclusion. No outliers were detected for activity estimates of both tasks, while four outlier values were removed from the analyses of the 2-back and 0-back performance.

The main analyses included the activity estimates modulated by light illuminance as a dependent variable and the hypothalamus subpart and stimulus type (2-back/0-back - neutral/emotional) as repeated measures (autoregressive (1) correlation), together with age, sex and BMI as covariates. The second set of post hoc GLMM analyses included the activity estimates of the hypothalamus subparts as the dependent variable and hypothalamus subpart, stimulus type and illuminance (0, 0.16, 37, 92, 190 mel EDI lux) as the repeated measures (autoregressive (1) correlation), together with age, sex, and BMI as covariates and interaction term between illuminance and hypothalamus subpart. The final set of analyses included performance metrics as dependent variables (accuracy to the 2-back or 0-back task - as percentage of correct responses; reaction time – ms - to emotional or neutral stimuli during the emotional task) and included the same repeated measures and covariates ads in the preceding set as well as activity of the relevant hypothalamus subpart.

Optimal sensitivity and power analyses in GLMMs remain under investigation (e.g. *Kain et al., 2015*). We nevertheless computed a prior sensitivity analysis to get an indication of the minimum detectable effect size in our main analyses given our sample size. According to G*Power 3 (version 3.1.9.4; *Faul et al., 2009*), taking into account a power of 0.8, an error rate $\alpha$ of 0.025 (correcting for 2 tasks), and a sample of 26 allowed us to detect large effect sizes $r>0.54$ (two-sided; absolute values; CI: 0.19–0.77; $R^2>0.29$, $R^2$ CI: 0.04–0.59) within a multiple linear regression framework including one tested predictor (illuminance effect) and three covariates (age, sex, and BMI).

## Acknowledgements

The study was conducted at the GIGA-In Vivo Imaging technological platform of ULiège, Belgium. The authors thank Christine Bastin, Annick Claes, Christian Degueldre, Catherine Hagelstein, Gregory Hammad, Brigitte Herbillon, Patrick Hawotte, Sophie Laloux, Erik Lambot, Benjamin Lauricella, André Luxen, Pierre Maquet, and Eric Salmon for their help over the different steps of the study. It was supported by the Belgian Fonds National de la Recherche Scientifique (FNRS; CDR J.0222.20), the European Union's Horizon 2020 research and innovation program under the Marie Skłodowska-Curie grant agreement No 860613, the Fondation Léon Frédéricq, ULiège - U Maastricht Imaging Valley, ULiège-Valeo Innovation Chair "Health and Well-Being in Transport" and Safran (LIGHT-CABIN), the European Regional Development Fund (Biomed-Hub), and Siemens. None of these funding sources had any impact on the design of the study nor the interpretation of the findings. AB is supported by Synergia Medial SA and the Walloon Region (Industrial Doctorate Program, convention n°8193). EB is supported by the Maastricht University - Liège University Imaging Valley. RS and FB are supported by the European Union's Horizon 2020 research and innovation program under the Marie Skłodowska-Curie grant agreement No 860613. IC, EK, IP, NM, and GV are supported by the FRS-FNRS. SS was supported by ULiège-Valeo Innovation Chair and Siemens Healthineers. PT and LL are supported by the EU Joint Programme Neurodegenerative Disease Research (JPND) IRONSLEEP and SCAIFIELD projects, respectively – FNRS references: PINT-MULTI R.8011.21 & 8006.20.

## Additional information

### Funding

| Funder | Grant reference number | Author |
| --- | --- | --- |
| Fonds De La Recherche Scientifique - FNRS Belgium | J.0222.20 | Gilles Vandewalle |
| Marie Sklodowska-Curie Actions | 10.3030/860613 | Roya Sharifpour<br>Jose Balda Aizpurua<br>Gilles Vandewalle |
| Fondation Léon Frédéricq | | Islay Campbell |
| ULiège - U. Maastricht Imaging Valley | | Elise Beckers |

| Funder | Grant reference number | Author |
|---|---|---|
| ULiège-Valeo Innovation Chair "Health and Well-Being in Transport" and Sanfran | LIGHT-CABIN | Siya Sherif<br>Gilles Vandewalle |
| European Regional Development Fund | Biomed Hub | Fabienne Collette<br>Christophe Phillips<br>Gilles Vandewalle |
| JPND - EU Joint-Programme Neurodegenerative Disease Research | SCAIFIELD / R.8011.21 | Laurent Lamalle<br>Gilles Vandewalle |
| JPND - EU Joint-Programme Neurodegenerative Disease Research | IRONSLEEP / R.8006.20 | Puneet Talwar<br>Gilles Vandewalle |

The funders had no role in study design, data collection and interpretation, or the decision to submit the work for publication.

## Author contributions

Islay Campbell, Conceptualization, Data curation, Formal analysis, Investigation, Methodology, Writing - original draft; Roya Sharifpour, Data curation, Formal analysis, Investigation, Methodology, Writing - original draft; Jose Fermin Balda Aizpurua, Elise Beckers, Ilenia Paparella, Alexandre Berger, Ekaterina Koshmanova, Nasrin Mortazavi, Data curation, Formal analysis, Writing – review and editing; John Read, Resources, Project administration, Writing – review and editing; Mikhail Zubkov, Christophe Phillips, Laurent Lamalle, Resources, Software, Validation, Writing – review and editing; Puneet Talwar, Siya Sherif, Resources, Software, Writing – review and editing; Fabienne Collette, Funding acquisition, Project administration, Writing – review and editing; Gilles Vandewalle, Conceptualization, Supervision, Investigation, Methodology, Writing - original draft, Project administration, Writing – review and editing

## Author ORCIDs

Puneet Talwar ⓘ https://orcid.org/0000-0001-7631-7926
Christophe Phillips ⓘ https://orcid.org/0000-0002-4990-425X
Laurent Lamalle ⓘ https://orcid.org/0000-0002-8930-7337
Gilles Vandewalle ⓘ https://orcid.org/0000-0003-2483-2752

## Ethics

Human subjects: The protocol was approved by the Ethics Committee of the Faculty of Medicine at the University of Liège (ref 2020/11). Participants gave their written informed consent to take part in the study and received monetary compensation for their participation.

Reviewer #1 (Public Review): https://doi.org/10.7554/eLife.96576.3.sa1
Reviewer #2 (Public Review): https://doi.org/10.7554/eLife.96576.3.sa2
Reviewer #3 (Public Review): https://doi.org/10.7554/eLife.96576.3.sa3
Author response https://doi.org/10.7554/eLife.96576.3.sa4

---

# Additional files

## Supplementary files

• Supplementary file 1. Online supplementary information. (a) Demographics of study sample. (b) Light characteristics. (c) Post hoc contrasts between illuminances within each hypothalamus subpart during the executive task. (d) Post hoc contrasts between illuminances within each hypothalamus subpart during the emotional task. (e) Post hoc contrasts between hypothalamus subpart for each illuminance during the executive task. (f) Post hoc contrasts between hypothalamus subpart for each illuminance during the emotional task. (g) Association between performance to the 2-back task and the activity of each hypothalamus subpart during each illuminance.

• MDAR checklist

## Data availability

The processed version of the data (consisting of the contrast of interest maps in MNI space) are available on ULiège Open Data Repository. The data tables and analysis scripts supporting the results included in this manuscript are publicly available via GitLab (copy archived at *Campbell et al., 2024b*). We used Matlab and Python scripts for MRI data processing and to compute performance metrics, while we used SAS for statistical analyses. The data cannot be used for commercial purposes without prior agreement of the corresponding author. The raw data could be identified and linked to a single subject and represent a huge amount of data (>200 Gb). Researchers willing to access to the raw should send a request to the corresponding author (GV). Data sharing will require evaluation of the request by the local Research Ethics Board and the signature of a data transfer agreement (DTA).

The following dataset was generated:

| Author(s) | Year | Dataset title | Dataset URL | Database and Identifier |
|---|---|---|---|---|
| Islay C, Roya S, Fermin BA, Elise B, Ilenia P, Alexandre B, Ekaterina K, John R, Mikhail Z, Puneet T, Fabienne C, Siya S, Christophe P, Laurent L, Gilles V, Nasrin M | 2024 | Replication Data for: 'Regional response to light illuminance across the human hypothalamus' | https://doi.org/10.58119/ULG/NXDMYW | ULiège Open Data Repository, 10.58119/ULG/NXDMYW |

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
