## [Editor Report · eLife assessment]

This **fundamental** work describes the complex interplay between light exposure, hypothalamic activity, and cognitive function. The evidence supporting the conclusion is **compelling** with potential therapeutic applications of light modulation. The work will be of broad interest to basic and clinical neuroscientists.

---

## [Referee Report · Reviewer #1 (Public Review)]

Summary:

Campbell et al investigated the effects of light on the human brain, in particular the subcortical part hypothalamus during auditory cognitive tasks. The mechanisms and neuronal circuits underlying light effects in non-image forming responses are so far mostly studied in rodents but are not easily translated in humans. Therefore, this is a fundamental study aiming to establish the impact light illuminance has on the subcortical structures using the high-resolution 7T fMRI. The authors found that parts of the hypothalamus are differently responding to illuminance. In particular, they found that the activity of the posterior hypothalamus increases while the activity of the anterior and ventral parts of the hypothalamus decreases under high illuminance. The authors also report that the performance of the 2-back executive task was significantly better in higher illuminance conditions. However, it seems that the activity of the posterior hypothalamus subpart is negatively related to the performance of the executive task, implying that it is unlikely that this part of the hypothalamus is directly involved in the positive impact of light on performance observed. Interestingly, the activity of the posterior hypothalamus was, however, associated with an increased behavioural response to emotional stimuli. This suggests that the role of this posterior part of the hypothalamus is not as simple regarding light effects on cognitive and emotional responses. This study is a fundamental step towards our better understanding of the mechanisms underlying light effects on cognition and consequently optimising lighting standards.

Strengths:

While it is still impossible to distinguish individual hypothalamic nuclei, even with the high-resolution fMRI, the authors split the hypothalamus into five areas encompassing five groups of hypothalamic nuclei. This allowed them to reveal that different parts of the hypothalamus respond differently to an increase in illuminance. They found that higher illuminance increased the activity of the posterior part of the hypothalamus encompassing the MB and parts of the LH and TMN, while decreasing the activity of the anterior parts encompassing the SCN and another part of TMN. These findings are somewhat in line with studies in animals. It was shown that parts of the hypothalamus such as SCN, LH, and PVN receive direct retinal input in particular from ipRGCs. Also, acute chemogenetic activation of ipRGCs was shown to induce activation of LH and also increased arousal in mice.

Weaknesses:

While the light characteristics are well documented and EDI calculated for all of the photoreceptors, it is not very clear why these irradiances and spectra were chosen. It would be helpful if the authors explained the logic behind the four chosen light conditions tested. Also, the lights chosen have cone-opic EDI values in a high correlation with the melanopic EDI, therefore we can't distinguish if the effects seen here are driven by melanopsin and/or other photoreceptors. In order to provide a more mechanistic insight into the light-driven effects on cognition ideally one would use silent substitution approach to distinguish between different photoreceptors. This may be something to consider when designing the follow-up studies.

---

## [Referee Report · Reviewer #2 (Public Review)]

Summary

The interplay between environmental factors and cognitive performance has been a focal point of neuroscientific research, with illuminance emerging as a significant variable of interest. The hypothalamus, a brain region integral to regulating circadian rhythms, sleep, and alertness, has been posited to mediate the effects of light exposure on cognitive functions. Previous studies have highlighted the role of the hypothalamus in orchestrating bodily responses to light, implicating specific neural pathways such as the orexin and histamine systems, which are crucial for maintaining wakefulness and processing environmental cues. Despite advancements in our understanding, the specific mechanisms through which varying levels of light exposure influence hypothalamic activity and, in turn, cognitive performance, remain inadequately explored. This gap in knowledge underscores the need for high-resolution investigations that can dissect the nuanced impacts of illuminance on different hypothalamic regions. Utilizing state-of-the-art 7 Tesla functional magnetic resonance imaging (fMRI), the present study aims to elucidate the differential effects of light on hypothalamic dynamics and establish a link between regional hypothalamic activity and cognitive outcomes in healthy young adults. By shedding light on these complex interactions, this research endeavours to contribute to the foundational knowledge necessary for developing innovative therapeutic strategies aimed at enhancing cognitive function through environmental modulation.

Strengths:

(1) Considerable Sample Size and Detailed Analysis: The study leverages a robust sample size and conducts a thorough analysis of hypothalamic dynamics, which enhances the reliability and depth of the findings.

(2) Use of High-Resolution Imaging: Utilizing 7 Tesla fMRI to analyze brain activity during cognitive tasks offers high-resolution insights into the differential effects of illuminance on hypothalamic activity, showcasing the methodological rigour of the study.

(3) Novel Insights into Illuminance Effects: The manuscript reveals new understandings of how different regions of the hypothalamus respond to varying illuminance levels, contributing valuable knowledge to the field.

(4) Exploration of Potential Therapeutic Applications: Discussing the potential therapeutic applications of light modulation based on the findings suggests practical implications and future research directions.

The current version of the manuscript addresses previous weaknesses, including details about the illuminance levels, light spectral characteristics used in the MRI study, and light patterns during behavioural tasks. The authors effectively tackle open questions in the field and provide solid evidence that enhances our understanding of the mechanisms underlying the effects of light on cognition.

---

## [Referee Report · Reviewer #3 (Public Review)]

Summary:

Campbell and colleagues use a combination of high-resolution fMRI, cognitive tasks and different intensities of light illumination to test the hypothesis that the intensity of illumination differentially impacts hypothalamic substructures that, in turn, promote alterations in arousal that affect cognitive and affective performance. The authors find evidence in support of a posterior-to-anterior gradient of increased blood flow in the hypothalamus during task performance that they later relate to performance on two different tasks. The results provide an enticing link between light levels, hypothalamic activity and cognitive/affective function, however clarification of some methodological choices will help to improve confidence in the findings.

Strengths:

* The authors' focus on the hypothalamus and its relationship to light intensity is an important and understudied question in neuroscience.

Weaknesses:

* I found it challenging to relate the authors hypotheses, which I found to be quite compelling, to the apparatus used to test the hypotheses - namely, the use of orange light vs. different light intensities; and the specific choice of the executive and emotional tasks, which differed in key features (e.g., block-related vs. event-related designs) that were orthogonal to the psychological constructs being challenged in each task.

* Given the small size of the hypothalamus and the irregular size of the hypothalamic parcels, I wondered whether a more data-driven examination of the hypothalamic time series would have provided a more parsimonious test of their hypothesis.

---

## [Author Response]

The following is the authors’ response to the original reviews.

**Reviewer #1 (Public Review):**
Summary:Campbell et al investigated the effects of light on the human brain, in particular the subcortical part of the hypothalamus during auditory cognitive tasks. The mechanisms and neuronal circuits underlying light effects in non-image forming responses are so far mostly studied in rodents but are not easily translated in humans. Therefore, this is a fundamental study aiming to establish the impact light illuminance has on the subcortical structures using the high-resolution 7T fMRI. The authors found that parts of the hypothalamus are differently responding to illuminance. In particular, they found that the activity of the posterior hypothalamus increases while the activity of the anterior and ventral parts of the hypothalamus decreases under high illuminance. The authors also report that the performance of the 2-back executive task was significantly better in higher illuminance conditions. However, it seems that the activity of the posterior hypothalamus subpart is negatively related to the performance of the executive task, implying that it is unlikely that this part of the hypothalamus is directly involved in the positive impact of light on performance observed. Interestingly, the activity of the posterior hypothalamus was, however, associated with an increased behavioural response to emotional stimuli. This suggests that the role of this posterior part of the hypothalamus is not as simple regarding light effects on cognitive and emotional responses. This study is a fundamental step towards our better understanding of the mechanisms underlying light effects on cognition and consequently optimising lighting standards.Strengths:While it is still impossible to distinguish individual hypothalamic nuclei, even with the highresolution fMRI, the authors split the hypothalamus into five areas encompassing five groups of hypothalamic nuclei. This allowed them to reveal that different parts of the hypothalamus respond differently to an increase in illuminance. They found that higher illuminance increased the activity of the posterior part of the hypothalamus encompassing the MB and parts of the LH and TMN, while decreasing the activity of the anterior parts encompassing the SCN and another part of TMN. These findings are somewhat in line with studies in animals. It was shown that parts of the hypothalamus such as SCN, LH, and PVN receive direct retinal input in particular from ipRGCs. Also, acute chemogenetic activation of ipRGCs was shown to induce activation of LH and also increased arousal in mice.Weaknesses:While the light characteristics are well documented and EDI calculated for all of the photoreceptors, it is not very clear why these irradiances and spectra were chosen. It would be helpful if the authors explained the logic behind the four chosen light conditions tested. Also, the lights chosen have cone-opic EDI values in a high correlation with the melanopic EDI, therefore we can't distinguish if the effects seen here are driven by melanopsin and/or other photoreceptors. In order to provide a more mechanistic insight into the light-driven effects on cognition ideally one would use a silent substitution approach to distinguish between different photoreceptors. This may be something to consider when designing the follow-up studies.
**Reviewer #1 (Recommendations For The Authors):**
(1) As suggested in the public review more information regarding the reasons behind the chosen light condition is needed.While the light characteristics are well documented and EDI calculated for all of the photoreceptors, it is not very clear why these irradiances and spectra were chosen. It would be helpful if the authors explained the logic behind the four chosen light conditions tested. Also, the lights chosen have cone-opic EDI values in a high correlation with the melanopic EDI, therefore we can't distinguish if the effects seen here are driven by melanopsin or cone opsins. In order to provide a more mechanistic insight into the light-driven effects on cognition ideally one would use a silent substitution approach to distinguish between different photoreceptors.(2) In support of this work, it was shown in mice that acute activation of ipRGCs using chemogenetics induces c-fos in some of the hypothalamic brain areas discussed here including LH (Milosavljevic et al, 2016 Curr Biol). Another study to consider including in the discussion is by Sonoda et al 2020 Science, in which the authors showed that a subset of ipRGCs release GABA.(3) Figure 1 looks squashed, especially the axes. Also, Figure 2 looks somewhat blurry. I would suggest that the authors edit the figures to correct this.

We thank the reviewer for their positive comments and agree with the weaknesses they pointed out.

(1) The explanation regarding the choice of the illuminance is now included in the revised manuscript (PAGE 17): “Blue-enriched light illuminances were set according to the technical characteristics of the light source and to keep the overall photon flux similar to prior 3T MRI studies of our team (between ~1012 and 1014 ph/cm²/s) (Vandewalle et al., 2010, 2011). The orange light was introduced as a control visual stimulation for potential secondary whole-brain analyses. For the present region of interest analyses, we discarded colour differences between the light conditions and only considered illuminance as indexed by mel EDI lux. This constitutes a limitation of our study as it does not allow attributing the findings to a particular photoreceptor class.”

The revised discussion makes clear that these choices limit the interpretation about the photoreceptors involved (PAGES 12-13): “We based our rationale and part of our interpretations on ipRGC projections, which have been demonstrated in rodents to channel the NIF biological impact of light and incorporate the inputs from rods and cones with their intrinsic photosensitivity into a light signal that can impact the brain (Güler et al., 2008; Tri & Do, 2019). Given the polychromatic nature of the light we used, classical photoreceptors and their projections to visual brain areas are, however, very likely to have directly or indirectly contributed to the modulation by light of the regional activity of the hypothalamus.”

The discussion also points out the promises of silent substitution (PAGE 13): “Future human studies could isolate the contribution of each photoreceptor class to the impact of light on cognitive brain functions by manipulating prior light history (Chellappa et al., 2014) or through the use of silent substitutions between metameric light exposures (Viénot et al., 2012)”.

(2) We now refer to the studies by Milosavljevic et al. and Sonoda et al.

PAGE 9: “Our data may therefore be compatible with an increase in orexin release by the LH with increasing illuminance. In line with this assumption, chemoactivation of ipRGCs lead to increase c-fos production, a marker of cellular activation, over several nuclei of the hypothalamus, including the lateral hypothalamus (Milosavljevic et al., 2016). If this initial effect of light we observe over the posterior part of the hypothalamus was maintained over a longer period of exposure, this would stimulate cognition and maintain or increase alertness (Campbell et al., 2023) and may also be part of the mechanisms through which daytime light increases the amplitude in circadian variations of several physiological features (BanoOtalora et al., 2021; Dijk et al., 2012).”

PAGE 10: “Chemoactivation of ipRGCs in rodents led to an increase activity of the SCN, over the inferior anterior hypothalamus, but had no impact on the activity of the VLPO, over the superior anterior hypothalamus (Milosavljevic et al., 2016). How our findings fit with these fine-grained observations and whether there are species-specific differences in the responses to light over the different part of the hypothalamus remains to be established.”

PAGE 10: “In terms of chemical communication, these changes in activity could be the results of an inhibitory signal from a subclass of ipRGCs, potentially through the release aminobutyric acid (GABA), as a rodent study found that a subset of ipRGCs release GABA at brain targets including the SCN (and intergeniculate leaflet and ventral lateral geniculate nucleus), leading to a reduction in the ability of light to affect pupil size and circadian photoentrainment (Sonoda et al., 2020). Whatever the signalling of ipRGC, our finding over the anterior hypothalamus could correspond to a modification of GABA signalling of the SCN which has been reported to have excitatory properties, such that the BOLD signal changes we report may correspond to a reduction in excitation arising in part from the SCN (Albers et al., 2017).”

(3) Figures 1 and 2 were modified. We hope their quality is now satisfactory. We are willing to provide separate figures prior to publication of the Version of Record.

**Reviewer #2 (Public Review):**
SummaryThe interplay between environmental factors and cognitive performance has been a focal point of neuroscientific research, with illuminance emerging as a significant variable of interest. The hypothalamus, a brain region integral to regulating circadian rhythms, sleep, and alertness, has been posited to mediate the effects of light exposure on cognitive functions. Previous studies have illuminated the role of the hypothalamus in orchestrating bodily responses to light, implicating specific neural pathways such as the orexin and histamine systems, which are crucial for maintaining wakefulness and processing environmental cues. Despite advancements in our understanding, the specific mechanisms through which varying levels of light exposure influence hypothalamic activity and, in turn, cognitive performance, remain inadequately explored. This gap in knowledge underscores the need for high-resolution investigations that can dissect the nuanced impacts of illuminance on different hypothalamic regions. Utilizing state-of-the-art 7 Tesla functional magnetic resonance imaging (fMRI), the present study aims to elucidate the differential effects of light on the hypothalamic dynamics and establish a link between regional hypothalamic activity and cognitive outcomes in healthy young adults. By shedding light on these complex interactions, this research endeavours to contribute to the foundational knowledge necessary for developing innovative therapeutic strategies aimed at enhancing cognitive function through environmental modulation.Strengths:(1) Considerable Sample Size and Detailed Analysis: The study leverages a robust sample size and conducts a thorough analysis of hypothalamic dynamics, which enhances the reliability and depth of the findings.(2) Use of High-Resolution Imaging: Utilizing 7 Tesla fMRI to analyze brain activity during cognitive tasks offers high-resolution insights into the differential effects of illuminance on hypothalamic activity, showcasing the methodological rigor of the study.(3) Novel Insights into Illuminance Effects: The manuscript reveals new understandings of how different regions of the hypothalamus respond to varying illuminance levels, contributing valuable knowledge to the field.(4) Exploration of Potential Therapeutic Applications: Discussing the potential therapeutic applications of light modulation based on the findings suggests practical implications and future research directions.Weaknesses:(1) Foundation for Claims about Orexin and Histamine Systems: The manuscript needs to provide a clearer theoretical or empirical foundation for claims regarding the impact of light on the orexin and histamine systems in the abstract.(2) Inclusion of Cortical Correlates: While focused on the hypothalamus, the manuscript may benefit from discussing the role of cortical activation in cognitive performance, suggesting an opportunity to expand the scope of the manuscript.(3) Details of Light Exposure Control: More detailed information about how light exposure was controlled and standardized is needed to ensure the replicability and validity of the experimental conditions.(4) Rationale Behind Different Exposure Protocols: To clarify methodological choices, the manuscript should include more in-depth reasoning behind using different protocols of light exposure for executive and emotional tasks.
**Reviewer #2 (Recommendations For The Authors):**
Attention to English language precision and correction of typographical errors, such as "hypothalamic nuclei" instead of "hypothalamus nuclei," is necessary for enhancing the manuscript.

We thank the reviewer for recognising the interest and strength of our study.

(1) As detailed in the discussion, we do believe orexin and histamine are excellent candidates for mediating the results we report. As also pointing out, however, we are in no position to know which neurons, nuclei, neurotransmitter and neuromodulator underlie the results. The last sentence of the abstract (PAGE 2) was therefore removed as we agree the statement was too strong. We carefully reconsider the discussion and believe that no such overstatement was present.

(2) Hypothalamus nuclei are connected to multiple cortical (and subcortical) structures. The relevance of these projections will vary with the cognitive task considered. In addition, we have not yet considered the cortex in our analyses such that truly integrating cortical structures appears premature.

We nevertheless added the following short statement (PAGE 11): “Subcortical structures, and particularly those receiving direct retinal projections, including those of the hypothalamus, are likely to receive light illuminance signal first before passing on the light modulation to the cortical regions involved in the ongoing cognitive process (Campbell et al., 2023).”

(3) We now include the following as part of the method section (PAGES 16-17): “Illuminance and spectra could not be directly measured within the MRI scanner due to the ferromagnetic nature of measurement systems. The coil of the MRI and the light stand, together with the lighting system were therefore placed outside of the MR room to reproduce the experimental conditions of the in a completely dark room. A sensor was placed 2 cm away from the mirror of the coil that is mounted at eye level, i.e. where the eye of the first author of the paper would be positioned, to measure illuminance and spectra. The procedure was repeated 4 times for illuminance and twice for spectra and measurements were averaged. This procedure does not take into account interindividual variation in head size and orbit shape such that the reported illuminance levels may have varied slightly across subjects. The relative differences between illuminance are, however, very unlikely to vary substantially across participants such that statistics consisting of tests for the impact of relative differences in illuminance were not affected. The detailed values reported in Supplementary Table 2 were computed combining spectra and illuminance using the excel calculator associated with a published work (Lucas et al., 2014).”

(4) The explanation regarding the choice of the illuminance is now included in the revised manuscript (PAGE 17): “Blue-enriched light illuminances were set according to the technical characteristics of the light source and to keep the overall photon flux similar to prior 3T MRI studies of our team (between ~1012 and 1014 ph/cm²/s) (Vandewalle et al., 2010, 2011). The orange light was introduced as a control visual stimulation for potential secondary whole-brain analyses. For the present region of interest analyses, we discarded colour differences between the light conditions and only considered illuminance as indexed by mel EDI lux. This constitutes a limitation of our study as it does not allow attributing the findings to a particular photoreceptor class.”

(5) The manuscript was thoroughly rechecked, and we hope to have spotted all typos and language errors.

**Reviewer #3 (Public Review):**
Summary:Campbell and colleagues use a combination of high-resolution fMRI, cognitive tasks, and different intensities of light illumination to test the hypothesis that the intensity of illumination differentially impacts hypothalamic substructures that, in turn, promote alterations in arousal that affect cognitive and affective performance. The authors find evidence in support of a posterior-to-anterior gradient of increased blood flow in the hypothalamus during task performance that they later relate to performance on two different tasks. The results provide an enticing link between light levels, hypothalamic activity, and cognitive/affective function, however, clarification of some methodological choices will help to improve confidence in the findings.Strengths:* The authors' focus on the hypothalamus and its relationship to light intensity is an important and understudied question in neuroscience.Weaknesses:(1) I found it challenging to relate the authors' hypotheses, which I found to be quite compelling, to the apparatus used to test the hypotheses - namely, the use of orange light vs. different light intensities; and the specific choice of the executive and emotional tasks, which differed in key features (e.g., block-related vs. event-related designs) that were orthogonal to the psychological constructs being challenged in each task.(4) Given the small size of the hypothalamus and the irregular size of the hypothalamic parcels, I wondered whether a more data-driven examination of the hypothalamic time series would have provided a more parsimonious test of their hypothesis.
**Reviewer #3 (Recommendations For The Authors):**
(1) The authors may wish to explain the importance of the orange light condition in the early section of the results -- i.e., when they first present the task structure. As it stands, I don't have a good appreciation of why the orange light was included -- was it a control condition? And if the differences between the light conditions (e.g., the narrow- vs. wide-band of light) were indeed ignored by focussing on the illuminance levels, are there any potential issues that the authors could then mitigate against with further experiments/analyses?(2) Are there other explanations for why illuminance levels might improve cognitive performance? For instance, the capacity to more easily perceive the stimuli in an experiment could plausibly make it easier to complete a given task. If this is the case, can the authors conceptualise a way to rule out this hypothesis?(3) Did the authors control for the differences in the number of voxels in each hypothalamic subregion? Or perhaps consider estimating the variance across voxels within the larger parcels, to determine whether the mean time series was comparable to the time series of the smaller parcels?(4) An alternative strategy that would mitigate against the differences in the size of hypothalamic parcels would be to conduct analyses on the hypothalamus without parcellation, but instead using dimensionality reduction techniques to observe the natural spread of responses across the hypothalamus. From the authors' results, my intuition is that these analyses will lead to similar conclusions, albeit without any of the potential issues with respect to differently-sized parcels.

We thank the reviewer for acknowledging the originality and interest of our study. We agree that some methodological choices needed more explanation. We will address the weaknesses they pointed out as follows:

(1) The explanation regarding the choice of the illuminance is now included in the revised manuscript (PAGE 17): “Blue-enriched light illuminances were set according to the technical characteristics of the light source and to keep the overall photon flux similar to prior 3T MRI studies of our team (between ~1012 and 1014 ph/cm²/s) (Vandewalle et al., 2010, 2011). The orange light was introduced as a control visual stimulation for potential secondary whole-brain analyses. For the present region of interest analyses, we discarded colour differences between the light conditions and only considered illuminance as indexed by mel EDI lux. This constitutes a limitation of our study as it does not allow attributing the findings to a particular photoreceptor class.”

The revised discussion makes clear that these choices limit the interpretation about the photoreceptors involved (PAGE 12-13): “We based our rationale and part of our interpretations on ipRGC projections, which have been demonstrated in rodents to channel the NIF biological impact of light and incorporate the inputs from rods and cones with their intrinsic photosensitivity into a light signal that can impact the brain (Güler et al., 2008; Tri & Do, 2019). Given the polychromatic nature of the light we used, classical photoreceptors and their projections to visual brain areas are, however, very likely to have directly or indirectly contributed to the modulation by light of the regional activity of the hypothalamus.”

We further mention that (PAGE 13): “Furthermore, we cannot exclude that colour and/or spectral differences between the orange and 3 blue-enriched light conditions may have contributed to our findings. Research in rodent model demonstrated that variation in the spectral composition of light was perceived by the suprachiasmatic nucleus to set circadian timing (Walmsley et al., 2015). No such demonstration has, however, been reported yet for the acute impact of light on alertness, attention, cognition or affective state.”

Regarding the choice of tasks, we added the following the method section (PAGE 18): “Prior work of our team showed that the n-back task and emotional task included in the present protocol were successful probes to demonstrate that light illuminance modulates cognitive activity, including within subcortical structures (though resolution did not allow precise isolation of nuclei or subparts) (e.g. (Vandewalle et al., 2007, 2010)). When taking the step of ultra-high-field imaging, we therefore opted for these tasks as our goal was to show that illuminance affects brain activity across cognitive domains while not testing for task-specific aspects of these domains.”

We further added to the discussion (PAGE 8): “The pattern of light-induced changes was consistent across an executive and an emotional task which consisted of block and an event-related fMRI design, respectively. This suggests that a robust anterior-posterior gradient of activity modulation by illuminance is present in hypothalamus across cognitive domains.”

(2) We are unsure what the reviewer refers to when he states that the experiment could make it easier to perceive a stimulus. Aside from the fact that illuminance can increase alertness and attention such that a stimulus may be better or more easily perceived/processed, we do not see how blocks of ambient light, i.e. a long-lasting visual stimulus, may render auditory stimulation (letters or pseudo-words in the present) easier to perceive. To our knowledge multimodal or cross-modal integration has been robustly demonstrated for short visual/auditory cues that would precede or accompany auditory/visual stimulation.

We are willing to clarify this issue in the text if we receive additional explanation from the reviewer.

(3) We added subpart size as covariate in the analyses (instead of subpart number) and it did not affect the output of the statistical analyses (Author response table 1).

For completeness, we further computed standard deviation of the activity estimates of the voxels within each parcel for the main analysis of the n-back tasks and found a main effect of subpart (Author response table 2) indicating that the variability of the estimates varied across subparts. Post hoc contrast and the display included in Author response image1 show however that the difference were not related to subpart size per see. It is in fact the largest subpart (subpart 4) that shows the largest variability while one of the smallest subpart (subpart 2) shows the lowest variability. Though it may have contributed, it is therefore unlikely to explain our findings. We consider the analyses reported in (Author response tables 1 and 2) and (Author response image 1) as very technical and did not include it in the supplementary material for conciseness. If the reviewer judges it essential, we can reconsider our decision.

While computing these analyses, we realized that there were errors in the table 1 reporting the statistical outcomes of the main analyses of the emotional task. The main statistical outputs remain the same except for a nominal main effect of the task (emotional vs. neutral) and the fact that post hoc show a consistent difference between the posterior subpart (subpart 3) and all the other subparts, rather than all the other subparts except for the difference with superior tubular hypothalamus subpart: p-corrected = 0.09. We apologise for this slight error and were unable to isolate its origin. It does not modify the rest of the analyses (which were also rechecked) and the interpretations.

**Author response table 1. sa4table1:** Recomputations of the main GLMMs using subpart sizes rather than subpart numbers as covariate of interest.

Executive task			
Main GLMM			
Effect	F value(df)	P value*	Partial R ^(2)
Subpart size	4.36,(4,200)	0.0021	0.08
Task	0.74,(1,25)	0.4	
Subpart size xtask type	0.68,(4,200)	0.61	
Age	0.33,(1,22)	0.57	
BMI	0.59,(1,22)	0.45	
Sex	0.01,(1,22)	0.91	
Emotional task			
Main GLMM			
Effect	F Value	P value*	Partial R ^(2)
Subpart size	9.64,(4,193)	< .0001	0.17
Task	4.55,(1,25)	0.043	0.15
Subpart size xstimulus type	0.41,(4,193)	0.8	
Age	0.4(1,22)	0.53	
BMI	0.02,(1,22)	0.89	
Sex	1.37,(1,22)	0.25	

**Author response image 1. sa4fig1:** Activity estimate variability per hypothalamus subpart and subpart size.

**Author response table 2. sa4table2:** Difference in activity estimate standard deviation between hypothalamus subparts during the n-back task.

Subpart number (GLMM)			
Effect	F value(df)	P value ^(**)	Partial R ^(2)
Hypothalamussubparts	32.3(4,260)	< 0.0001	0.33
Task	< 0.001(1,23)	0.97	
Hypothalamussubparts x tasktype	0.03(4,260)	0.99	
Post hocs			
subpart	T value	p	P
1 vs. 2	5.24	< .0001	< .0001
1 vs. 3	0.30	0.77	0.99
1 vs. 4	1.47	0.14	0.58
1 vs. 5	-5.92	< .0001	< .0001
2 vs. 3	-4.95	< .0001	< .0001
2 vs. 4	-3.77	0.0002	0.0019
2 vs. 5	-11.17	< .0001	< .0001
3 vs. 4	1.18	0.24	0.77
3 vs. 5	-6.22	< .0001	< .0001
4 vs. 5	-7.39	< .0001	< .0001

Outputs of the generalized linear mixed model (GLMM) with subject as the random factor (intercept and slope), and task and subpart as repeated measures (ar(1) autocorrelation).

* The corrected p-value for multiple comparisons over 2 tests is p < 0.025.

# Refer to Fig.2A for correspondence of subpart numbers

The text referring to Table 1 was modified accordingly (PAGE 5): “A nominal main effect of the task was detected for the emotional task [p = 0.049; Table 1] but not for the n-back task. For both tasks, there was no significant main effect for any of the other covariates and post hoc analyses showed that the index of the illuminance impact was consistently different in the posterior hypothalamus subpart compared to the other subparts [pcorrected ≤ 0.05]”.

(4) We agree that a data driven approach could have constituted an alternative means to tests our hypothesis. We opted for an approach that we mastered best, while still allowing to conclusively test for regional differences in activity across the hypothalamus. Examination of time series of the very same data we used will mainly confirm the results of our analyses – an anterior-posterior gradient in the impact of illuminance - while it may yield slight differences in the boarders of the subparts of the hypothalamus undergoing decreased or increased activity with increasing illuminance. While the suggested approach may have been envisaged if we had been facing negative results (i.e. no differences between subparts, potentially because subparts would not reflect functional differences in response to illuminance change), it would constitute a circular confirmation of our main findings (i.e. using the same data). While we truly appreciate the suggestion, we do not consider that it would constitute a more parsimonious test of our hypothesis, now that we successfully applied GLM/parcellation and GLMM approaches.

We added the following statement to the discussion to take this comment into account (PAGE 12): “Future research may consider data-driven analyses of hypothalamus voxels time series as an alternative to the parcellation approach we adopted here. This may refine the delineation of the subparts of the hypothalamus undergoing decreased or increased activity with increasing illuminance.”

Response references

Albers, H. E., Walton, J. C., Gamble, K. L., McNeill, J. K., & Hummer, D. L. (2017). The dynamics of GABA signaling: Revelations from the circadian pacemaker in the suprachiasmatic nucleus. Frontiers in Neuroendocrinology, 44, 35–82. https://doi.org/10.1016/J.YFRNE.2016.11.003

Bano-Otalora, B., Martial, F., Harding, C., Bechtold, D. A., Allen, A. E., Brown, T. M., Belle, M. D. C., & Lucas, R. J. (2021). Bright daytime light enhances circadian amplitude in a diurnal

mammal. Proceedings of the National Academy of Sciences of the United States of America, 118(22), e2100094118.https://doi.org/10.1073/PNAS.2100094118/SUPPL_FILE/PNAS.2100094118.SAPP.PDF

Campbell, I., Sharifpour, R., & Vandewalle, G. (2023). Light as a Modulator of Non-Image-Forming Brain Functions Positive and Negative Impacts of Increasing Light Availability. Clocks & Sleep, 5(1), 116. https://doi.org/10.3390/CLOCKSSLEEP5010012

Chellappa, S. L., Ly, J. Q. M., Meyer, C., Balteau, E., Degueldre, C., Luxen, A., Phillips, C., Cooper, H. M., & Vandewalle, G. (2014). Photic memory for executive brain responses. Proceedings of the National Academy of Sciences of the United States of America, 111(16), 6087–6091. https://doi.org/10.1073/pnas.1320005111

Dijk, D. J., Duffy, J. F., Silva, E. J., Shanahan, T. L., Boivin, D. B., & Czeisler, C. A. (2012). Amplitude reduction and phase shifts of melatonin, cortisol and other circadian rhythms after a gradual advance of sleep and light exposure in humans. PloS One, 7(2). https://doi.org/10.1371/JOURNAL.PONE.0030037

Güler, A. D., Ecker, J. L., Lall, G. S., Haq, S., Altimus, C. M., Liao, H. W., Barnard, A. R., Cahill, H., Badea, T. C., Zhao, H., Hankins, M. W., Berson, D. M., Lucas, R. J., Yau, K. W., & Hattar, S. (2008). Melanopsin cells are the principal conduits for rod-cone input to non-image-forming vision. Nature, 453(7191), 102–105. https://doi.org/10.1038/nature06829

Lucas, R. J., Peirson, S. N., Berson, D. M., Brown, T. M., Cooper, H. M., Czeisler, C. A., Figueiro, M. G., Gamlin, P. D., Lockley, S. W., O’Hagan, J. B., Price, L. L. A., Provencio, I., Skene, D. J., & Brainard, G. C. (2014). Measuring and using light in the melanopsin age. Trends in Neurosciences, 37(1), 1–9. https://doi.org/10.1016/j.tins.2013.10.004

Milosavljevic, N., Cehajic-Kapetanovic, J., Procyk, C. A., & Lucas, R. J. (2016). Chemogenetic Activation of Melanopsin Retinal Ganglion Cells Induces Signatures of Arousal and/or Anxiety in Mice. Current Biology, 26(17), 2358–2363. https://doi.org/10.1016/j.cub.2016.06.057

Sonoda, T., Li, J. Y., Hayes, N. W., Chan, J. C., Okabe, Y., Belin, S., Nawabi, H., & Schmidt, T. M. (2020). A noncanonical inhibitory circuit dampens behavioral sensitivity to light. Science (New York, N.Y.), 368(6490), 527–531. https://doi.org/10.1126/SCIENCE.AAY3152

Tri, M., & Do, H. (2019). Melanopsin and the Intrinsically Photosensitive Retinal Ganglion Cells: Biophysics to Behavior. Neuron, 104, 205–226. https://doi.org/10.1016/j.neuron.2019.07.016

Vandewalle, G., Hébert, M., Beaulieu, C., Richard, L., Daneault, V., Garon, M. Lou, Leblanc, J., Grandjean, D., Maquet, P., Schwartz, S., Dumont, M., Doyon, J., & Carrier, J. (2011). Abnormal hypothalamic response to light in seasonal affective disorder. Biological Psychiatry, 70(10), 954–961. https://doi.org/10.1016/j.biopsych.2011.06.022

Vandewalle, G., Schmidt, C., Albouy, G., Sterpenich, V., Darsaud, A., Rauchs, G., Berken, P. Y., Balteau, E., Dagueldre, C., Luxen, A., Maquet, P., & Dijk, D. J. (2007). Brain responses to violet, blue, and green monochromatic light exposures in humans: Prominent role of blue light and the brainstem. PLoS ONE, 2(11), e1247. https://doi.org/10.1371/journal.pone.0001247

Vandewalle, G., Schwartz, S., Grandjean, D., Wuillaume, C., Balteau, E., Degueldre, C., Schabus, M., Phillips, C., Luxen, A., Dijk, D. J., & Maquet, P. (2010). Spectral quality of light modulates emotional brain responses in humans. Proceedings of the National Academy of Sciences of the United States of America, 107(45), 19549–19554. https://doi.org/10.1073/pnas.1010180107

Viénot, F., Brettel, H., Dang, T.-V., & Le Rohellec, J. (2012). Domain of metamers exciting intrinsically photosensitive retinal ganglion cells (ipRGCs) and rods. Journal of the Optical Society of America A, 29(2), A366. https://doi.org/10.1364/josaa.29.00a366

Walmsley, L., Hanna, L., Mouland, J., Martial, F., West, A., Smedley, A. R., Bechtold, D. A., Webb, A. R., Lucas, R. J., & Brown, T. M. (2015). Colour As a Signal for Entraining the Mammalian Circadian Clock. PLOS Biology, 13(4), e1002127. https://doi.org/10.1371/journal.pbio.1002127